# Quantifying the sensitivity of aerosol optical properties to the parameterizations of physico-chemical processes during the 2010 Russian wildfires and heatwave

Laura Palacios-Peña[1], Philip Stier[2], Raquel Lorente-Plazas[3], and Pedro Jiménez-Guerrero[1,4]

[1]Physics of the Earth, Regional Campus of International Excellence "Campus Mare Nostrum", University of Murcia, Spain.
[2]Atmospheric, Oceanic and Planetary Physics, Department of Physics, University of Oxford, UK.
[3]Dept. of Meteorology, Meteored, Almendricos, Spain.
[4]Biomedical Research Institute of Murcia (IMIB-Arrixaca), Spain.

**Correspondence:** Pedro Jiménez-Guerrero (pedro.jimenezguerrero@um.es)

**Abstract.** The impact of aerosol-radiation and aerosol-clouds interactions on the radiative forcing is subject to large uncertainties. This is caused by the limited understanding of aerosol optical properties and the role of aerosols as cloud condensation/ice nuclei (CCN/IN). On the other hand, aerosol optical properties and vertical distribution are highly related and their uncertainties come from different processes. This work attempts to quantify the sensitivity of aerosol optical properties (i.e. aerosol optical depth; AOD) and their vertical distribution (using the extinction coefficient, backscatter coefficient, and concentrations species profiles) to key processes. In order to achieve this objective, sensitivity tests have been carried out, using the WRF-Chem regional fully-coupled model by modifying the dry deposition, sub-grid convective transport, relative humidity and wet scavenging. The 2010 Russian heatwave/wildfire episode has been selected as case study.

Results indicate that AOD is sensitive to these key processes in the following order of importance: 1) modification of relative humidity, causing AOD differences up to 0.6; 2) modification of vertical convection transport with AOD differences around -0.4; and 3) the dry deposition with AOD absolute differences up to -0.35 and 0.3. Moreover, these AOD changes exhibit a non-linear response. Both, an increase and a decrease in the RH result in higher AOD values. On the other hand, both, the increase and offset of the sub-grid convective transport lead to a reduction in the AOD over the fire area. In addition, a similar non-linear response is found when reducing the dry deposition velocity; in particular, for the accumulation mode where the concentration of several species increases (while a decrease might be expected). These non-linear responses are highly dependent on the equilibrium of the thermodynamics system sulphate-nitrate-SOA (secondary organic aerosol). In this sense, small changes in the concentration of one species can strongly affect others, finally affecting aerosol optical properties. Changes in this equilibrium could come from modifications in relative humidity, dry deposition or vertical convective transport. By itself, dry deposition also presents a high uncertainty influencing the AOD representation.

# 1 Introduction

Since the First Assessment Report of the Intergovernmental Panel on Climate Change (IPCC), a wide scientific consensus identifies atmospheric aerosols and clouds as one of the forcing agents with the largest uncertainty in the climate system (Charlson et al., 1992; Schimel et al., 1996; Penner et al., 2001; Randall et al., 2007; Forster et al., 2007; Boucher, 2015). Atmospheric aerosols modify the Earth's radiative budget through aerosol-radiation interactions (ARI) and aerosol-cloud interactions (ACI). ARI lead to a redistribution of radiative energy in the atmosphere through scattering and absorption. In addition, ACI modify cloud microphysical properties and precipitation regimes as well as cloud effects on radiation (Randall et al., 2007; Boucher et al., 2013).

ARI and ACI are strongly dependent on aerosol optical properties and the ability of aerosols to act as cloud condensation nuclei (CCN) or ice nuclei (IN), which are controlled by the spatio-temporal aerosol distribution, the aerosol size, composition and mixing state (Stier et al., 2005). Thus, to determine and constrain the uncertainty in aerosol optical properties is a key issue for a better assessment of the uncertainty in aerosol effects.

Numerical models are useful tools for understanding the different parameters influencing the atmospheric system, such as aerosol optical properties. The complexity of how aerosols are treated in models varies widely, since these models take into account processes as emission, transport, deposition, microphysics and chemistry (Kipling et al., 2016). Differences in complexity primarily arise from representations of aerosol size distribution and mixing states. The most complex and realistic models are those considering the inclusion of ARI and ACI since they allow a fully-coupled interaction of aerosols, meteorology, radiation and chemistry. One example of these numerical models is WRF-Chem (Grell et al., 2005), used in this work. Notwithstanding the complexity of aerosol treatment in these models, there are still high uncertainties in processes representing the aerosol optical properties.

As stated by previous works (e.g. Palacios-Peña et al., 2017, 2018, 2019a), uncertainties in aerosol optical properties may be influenced by a number of factors, namely emissions; aerosol mass concentration; particle size representation (Balzarini et al., 2015); vertical distribution and location with respect to other forcing agents as clouds (Kipling et al., 2016); dry deposition and CCN (Romakkaniemi et al., 2012; Lee et al., 2013; Forkel et al., 2015); relative humidity (RH; Yoon and Kim, 2006; Zhang et al., 2012; Altaratz et al., 2013; Weigum et al., 2016); and aerosol internal mixing rules (Curci et al., 2019; Zhang et al., 2012).

Precisely, aerosol vertical distribution is highly influenced by aerosol optical properties (Palacios-Peña et al., 2018, 2019a). Henceforth, Kipling et al. (2016) investigated the uncertainty in the vertical layering of aerosol particles to different parameters: convective transport, emissions injection and size; vertical advection, boundary-layer mixing, entrainment into convective plumes, condensation, coagulation, nucleation, aqueous chemistry, aging of insoluble particles, Aitken transition to accumulation mode, dry deposition, in-cloud and below-cloud scavenging and re-evaporation. The convective transport and the in-cloud scavenging were found to be very important when controlling the vertical profile of all-aerosol components by mass and those with the highest influence on aerosol optical depth (AOD; Kipling et al., 2016).

The representation of CCN has been also identified as another second-order source of uncertainty in aerosol optical properties, such as AOD. An increase in downward solar radiation was found by Forkel et al. (2015) and Romakkaniemi et al. (2012) when ACI were taken into account. This latter contribution found a relationship between a reduction in the AOD and CCN because the inclusion of ACI in numerical models leads to a reduction in CCN by the condensation kinetics of water during cloud droplet formation. This induces a reduction of the cloud droplet number, the cloud liquid water and, finally, an increase in downward solar radiation. In addition to AOD, CCN conditioned the uncertainty in ACI, as well as cloud occurrence and cloud-related processes (updraught speeds, precipitation processes, etc.). Because of that, the high uncertainty existing when modelling CCN was evaluated by Lee et al. (2013), finding that dry deposition was the most important process for this uncertainty over more than twenty-eight model parameters selected by expert elicitation, including nucleation, aerosol ageing, pH of cloud drops, nucleation scavenging, dry deposition, modal with mode separation diameter, emissions and production of secondary organic aerosols (SOA). These results, which are partly because wet deposition was not fully varied, were found in one model framework (with its own structural uncertainties).

Another source of uncertainty is the aerosol variability at scales smaller than the model's grid box, which can hamper the representation of aerosol optical properties. This fact was brought to light in Weigum et al. (2016), where the aerosol water uptake through aerosol-gas equilibrium reactions was established as one of the most affected processes by this variability. The inherent non-linearities in these processes result in large changes in aerosol properties which are exaggerated by convective transport. The uncertainties in RH also contribute to those of aerosol optical properties due to their dependence in hygroscopic growth (Yoon and Kim, 2006; Zhang et al., 2012; Altaratz et al., 2013; Palacios-Peña et al., 2019a).

Bearing in mind the uncertainties described above, the aim of this work is to shed some light on the uncertainties when representing aerosol optical properties. In order to achieve this aim, this contribution quantifies the sensitivity of aerosol optical properties and their vertical distribution (which may condition aerosol radiative forcing) to several aerosol processes and parameters. This quantification has been estimated by sensitivity tests carried out using the WRF-Chem regional fully-coupled model. Modified aerosol processes and parameters are dry deposition, sub-grid convective transport, relative humidity and wet scavenging.

## 2 Methodology

Sensitivity tests have been conducted to assess the impact of the most relevant processes for representing aerosol optical properties. For that, the WRF-Chem model (Grell et al., 2005) version 3.9.1.1 has been utilized. The 2010 Russian heatwave/wildfires episode has been selected as a case study because of the literature available referring to this episode (see section 2.1). To achieve this objective, aerosol dry deposition velocity, sub-grid convective transport, aerosol water uptake and wet scavenging were the processes scaled. The degree of impact of these processes is evaluated by analyzing the AOD at 550 nm, different vertical profiles for extinction ($\alpha$) and backscatter coefficient ($\beta$) at 532 nm, and the concentration profiles of different aerosol species. The AOD is defined as the vertical integral of extinction in the total atmospheric column.

## 2.1 The 2010 Russian wildfires and heatwave episode

The 2010 Russian wildfires and heatwave episode occurred approximately from 25 of July to 15 of August 2010 and lasted a total of 22 days. This was an anomalous heatwave, termed as "mega-heatwave" by Barriopedro et al. (2011), with a prolonged blocking anticyclone situation which favoured an increase of the summer temperature (close to 9 degrees larger than 2002-2009 summers) promoting to larger wildfires (Bondur, 2011). This prolonged blocking situation has been attributed to the global warming leading to very high sea surface temperatures in several places around the world, due to the action of the ENSO (El Niño Southern Oscillation) which altered the atmospheric circulation by forcing quasi-stationary Rossby waves (Sedlàček et al., 2011; Lau and Kim, 2012; Trenberth and Fasullo, 2012). In addition, according to Rahmstorf and Coumou (2011) the 2010 July heat record in Moscow was caused by the climate warming with approximate 80 % probability.

With respect to air quality, this is a well-known and widely studied episode. Many of these works analyzed the physico-chemical characteristics of the smoke from wildfires and the effects on air quality of the transport (both particles and trace gases) to surrounding areas (Zvyagintsev et al., 2011; Witte et al., 2011; van Donkelaar et al., 2011; Gorchakov et al., 2014; Safronov et al., 2015); medium-range transport (e.g. Finland) (Portin et al., 2012; Mielonen et al., 2013) or long-range transport, even reaching Greece (Diapouli et al., 2014).

Among all these reasons, this heatwave has been extensively investigated because of the particularly significant interactions between meteorology and chemistry/particles during this strong air pollution episode (Makar et al., 2015b, a; Kong et al., 2015). This episode was one of the case studies within the COST Action ES1004 EuMetChem (European framework for online integrated air quality and meteorology modelling; see http://www.eumetchem.info/) chosen from the previous experience of Phase 2 of the Air Quality Modelling Evaluation International Initiative (AQMEII; Galmarini et al., 2015).

The effects of air pollution on meteorology where evinced by Konovalov et al. (2011), Chubarova et al. (2012) and Wong et al. (2012) among others. These studies demonstrated changes in atmospheric regional conditions caused by a modification in the composition of atmospheric gases; and also because of changes in optical and radiative characteristics of aerosols coming from the fire emissions. Gorchakov et al. (2014) detected a regional mean AOD of $1.02 \pm 0.02$ and a single-scattering albedo of 0.95; and estimated a regional mean aerosol radiative forcing at the top and the bottom of the atmosphere of $-61 \pm 1$ and $-107 \pm 2$ W m$^{-2}$, respectively.

When aerosol interactions were taken into account, a reduction of solar radiation on the ground up to 50 W m$^{-2}$ in diurnal averages and in the near-surface air temperature between 0.2 and 2.6 °K was evaluated on a regional scale over most of eastern Europe. Similarly, a reduction in the planetary boundary layer (PBL) height from 13 to 65 % and the vertical wind speed from 5 to 80 % were found by Péré et al. (2014). Baró et al. (2017) reported similar results on surface winds caused by a decrease of the shortwave downwelling radiation at the surface, leading to a reduction of the 2-m temperature and hence reducing the turbulent flux and developing a more stable PBL. This cooling increases both the surface pressure over the Russian area and the RH (with values around +3.5 %). In the same case, Forkel et al. (2016) manifested a reduction between 10 and 100 W m$^{-2}$ in the average downward short-wave radiation at the ground level and a drop in the mean 2-m temperature of almost 1 °K over the area where the fires took place. On the other hand, Péré et al. (2015) evaluated the impact of aerosol solar extinction on the

photochemistry, resulting in a reduction of the photolysis rates of $NO_2$ and $O_3$ up to 50 % (daytime average) due to the aerosol extinction along the aerosol plume transported, as well as a reduction of the formation of secondary aerosols.

## 2.2 Model setup

As aforementioned, the version 3.9.1.1 of the fully-coupled on-line WRF-Chem model (Grell et al., 2005) was used in order to simulate transport, mixing, and chemical transformation of trace gases and aerosols coupled to the meteorology (thus including

ARI and ACI processes, among others).

Figure 1 displays the target domain of the simulations which covered Europe with a horizontal resolution of $\sim 23\,km$. However, in order to focus on the aerosol effects, a smaller window covering between 40 and 65 $^\circ$ N and 20 and 60 $^\circ$ E (green box in Figure 1) was defined.

The definition of the modelling domain, initial and boundary meteorological and chemical conditions and different emissions

has been built on the previous experiences of the COST Action EuMetChem and Phase 2 of the AQMEII initiative. However, in this case the simulations are continuous runs instead of reinitialized every 48 hours (two-day time slices) as done in AQMEII and EuMetChem methodologies (Forkel et al., 2015). A spin-up period of five days has been considered for running the sensitivity tests.

Meteorological initial and boundary conditions (3-hourly data and $0.25^\circ$ resolution) were provided by the European Cen-

tre for Medium-Range Weather Forecasts (ECMWF) operational archive. Chemistry boundary conditions (3-hourly data and $1.125^\circ$ resolution) for the main trace gases and particulate matter concentrations were taken from the ECMWF Integrated Forecasting System – Model for Ozone and Related Chemical Tracers (IFS-MOZART) model run in the (MACC-II) project (Monitoring Atmospheric Composition and Climate-Interim Implementation; Inness et al., 2013).

Annual anthropogenic emission ($\sim 7\,km$ resolution), whose details are described in Im et al. (2015a, b), came from the

Netherlands Organization for Applied Scientific Research (TNO) MACC emissions inventory (http://www.gmes-atmosphere. eu/; Pouliot et al., 2012; Kuenen et al., 2014; Pouliot et al., 2015). $CH_4$, CO, $NH_3$, total Non-Methane Volatile Organic Compounds (NMVOCs), $NO_x$, PM ($PM_{10}$ and $PM_{2.5}$) and $SO_2$ were available by 10 activity sectors. Schaap et al. (2005) provided temporal (diurnal, day-of-week, seasonal) and vertical emission profiles. Biomass burning emission data of the total PM emissions (daily data with a spatial resolution of $0.1\,^\circ$) were derived from the project IS4FIRES (Integrated monitoring

and modelling system for wild-land fires; Sofiev et al., 2009). As described by Soares et al. (2015) emissions were calculated from a re-analysis of the fire radiative power from MODIS on-board of Aqua and Terra satellites and calibration emission factors based on the comparison between observations and modelled data processed by the System for Integrated modeLing of Atmospheric coMposition (SILAM). Day and night vertical injection profiles were also provided. Finally, total PM emissions were transformed to WRF-Chem emission species following Andreae and Merlet (2001) and and Wiedinmyer et al. (2011).

No heat release due to the fires was considered. Uncertainties of this biomass burning emissions dataset were estimated by Soares et al. (2015) with an overestimation in-average of 20–30 % which could raise to about 50 % in specific episodes. This impacts on total emissions likely come from under-stated injection height which can lead to overestimation of the near-surface concentration and reduction of elevated plumes; or a misinterpretation by MODIS of oil and gas flares and large

industrial installation as fires. More details can be found in Soares et al. (2015). Table 1 summarizes the physico-chemical

parameterizations and schemes used in the simulations.

The skills of the model to represent AOD during this episode have been evaluated in depth in Palacios-Peña et al. (2018) and Palacios-Peña et al. (2019a). The model skillfully represents low and mean AOD values albeit underestimates the high AOD over the Russian area due to two different hypothesis: 1) not considering the fire emissions from small fires (Toll et al., 2015; Wooster et al., 2005) or 2) a misrepresentation of the aerosol vertical profile based on the understated injection height of the

total biomass burning emissions found by Soares et al. (2015).

## 2.3 Sensitivity tests

Table 2 summarizes the sensitivity tests carried out. As previously mentioned, the processes selected to be scaled include RH, dry deposition, sub-grid convective transport and wet scavenging. They were chosen because they are considered as key sources of uncertainty when modelling atmospheric aerosol properties and thus they are expected to impact the estimation of aerosol

optical properties (e.g. Ackermann et al. (1998); Lee et al. (2013); Quan et al. (2016), among many others).

RH highly impacts aerosol properties by affecting several processes such as nucleation, chemistry or uptake of water through aerosol-gas equilibrium reactions (Ackermann et al., 1998). Because of that, our sensitivity test for this variable modified the RH in the aerosol module of WRF-Chem (precisely, in the part of the code when RH enters the aerosol module). Henceforth, RH modification only affects aerosol properties and not meteorology. Following the evaluation of this meteorological variable

conducted by Tuccella et al. (2012) and Žabkar et al. (2015), it was scaled to 0.9 (a reduction of 10 %). Although the translation into saturation only applies at saturation conditions, supersaturation values higher than 1 % are unlikely. Because of that, this variable could not be scaled by +10% (to 1.1), and hence the chosen upper values were 1.005 and 1.01; that is, 0.5 % and 1 % supersaturation respectively.

In this work, dry deposition velocity (DDV) is estimated by the MADE module (Ackermann et al., 1998) as in the Regional

Particulate Model (RPM; Binkowski and Shankar, 1995). But in contrast to RPM, MADE calculates and applies deposition velocities separately for each mode (Aitken, accumulation and coarse). The method uses the aerodynamic resistance, the settling velocity and Brownian diffusivity; and then, the Slinn and Slinn (1980)'s and Pleim et al. (1984)'s expressions are calculated by averaging the quantities over the $k^{th}$ moment of the distribution as in Kramm et al. (1992). The modification for our sensitivity test regarding dry deposition consists on scaled DDV by the values indicated in Table 2. Following Lee

et al. (2013), DDV has been scaled to 0.5 and 2 for the Aitken mode and 0.1 and 10 for the Accumulation mode, which are the both ends of the uncertainty range of these parameters. WRF-Chem configuration gives the opportunity to turn on/off the dry deposition of gases and aerosols. Thus, another sensitivity case corresponds to the WRF-Chem configuration with the dry deposition of aerosol turned off (aer_drydep_opt = 0 in the namelist of the model).

Analogously to dry deposition, sub-grid convective transport in WRF-Chem can be turned on/off. This process is parametrized

by a simple scheme (Grell and Dévényi, 2002) based on a convective parametrization developed by Grell (1993) and Grell et al. (1994). This scheme estimates the output temporal tendency (s $^{-1}$) separately in the bottom layer and the rest of the layers. Afterwards this tendency is applied to the chemical concentration for each species in order to estimate the sub-grid convective

transport. This tendency has been modified in our sensitivity test as indicated in Table 2. Following the evaluations carried out by Doherty et al. (2005) and Quan et al. (2016), the output temporal tendency has been scaled to $\pm 50\%$. Moreover, a case with sub-grid convective transport turned off (chem_conv_tr = 0 in the model's namelist) has been run.

Aerosol wet scavenging in WRF-Chem follows the approach of Easter et al. (2004). This process is produced by impacting/interception and precipitation, when all aerosol species are assumed to be immediately wet-deposited. The model distinguishes between wet scavenging for large-scale and sub-grid stratiform and sub-grid convective clouds. Both, stratiform (wetscav_onoff) and convective (conv_tr_wetscav) wet scavenging can be turned on/off in WRF-Chem. A case in which stratiform wet scavenging is turned off was run. This modification has been chosen because the evaluated episode was an anticyclonic situation without important convective clouds.

## 3  Results and discussion

In this section the results of the sensitivity of AOD representation to changes in RH, DDV, wet scavenging and convective transport are assessed, focusing on the Russian region affected by the heatwave-wildfire episode. Afterwards, a local evaluation of the vertical profiles is carried out in order to establish the influence of each process on aerosol vertical profiles.

### 3.1  Changes in total AOD

Figure 2,top-right displays the modelled AOD at 550 nm for the base case. The rest of the Figure 2 depicts the differences between the sensitivity experiments and the base case. For the base case, high AOD values (up 0.5) are found over a large area of central Russia, including populated cities such as Moscow, Nizhny Novgorod or Kazan. AOD values around 0.3, are found over a wider area close to the Finnish border (northwest of the domain) and over most of Belarus, Ukraine and the Black Sea (south of the domain). The lowest values (around 0.1) are found over central Europe. The changes of AOD in the sensitivity experiments are shown in the other pannels of Figure 2

Figure 2,*a*, *b* and *c* represent the sensitivity to RH: a decrease of 10% (L10RH); an increase of 0.5% (H05RH); and an increase of 1% (H1RH), respectively. As expected, a 10% decrease of the RH leads to a stronger response compared with the experiments when RH increases since the percentage of modification is lower in the latter sensitivity tests. L10RH (Figure 2,*a*) experiment shows positive differences at the west of the Volga river, reaching values around +0.6. Oppositely, there are negative differences of -0.15 in the area placed eastern to the Volga. Meanwhile, the H05RH (Figure 2,*b*) experiment shows this positive (west)/negative (east) dipole over the fire-affected area but differences are lower than 0.15.The H1RH experiment (Figure 2,*c*) promotes an increase of AOD encompassing most of the fire-affected area with values around +0.2.

Figure 2,*d* stands for the No-Dry Deposition case (NO_DD): Figure 2,*e* and *f* are the experiments with Low and High Dry Deposition for the Aitken mode, respectively (LDDV_AIT and HDDV_AIT); and Figure 2,*g* and *h* represent the tests modifying the Accumulation mode (LDDV_ACC and HDDV_ACC). All the experiments related to changes in dry deposition (Figure 2,*d-h*) showed its strongest response located over the wildfires area, but this response is less relevant than for other cases. Figures 2,*d*, NO_DD, and *e*, LDDV_AIT, have a similar spatial pattern of differences with positive changes (up to +0.35

and +0.2, respectively) at the western Volga river. However, increasing the dry deposition in Aitken mode (Figure 2,*f*) and both increasing and decreasing the deposition in accumulation mode (Figures 2,*g* and *h*) provoke negative changes of AOD over the eastern Volga (around -0.3 in all of these cases). HDDV_ACC is the only test which produces a general reduction in AOD over most of the study area (temporal and spatial mean change in AOD of -0.06) but the differences are stronger over fire affected areas and downwind.

Figure 2,*i* shows the No sub-grid Convective Transport (NO_CONV_TR) case and Figure 2,*k* the High sub-grid Convective Transport (HCONV_TR) case. Both of them evidence negative differences (up to -0.39 and -0.43, respectively) over the fire-affected and downwind areas. However, the NO_CONV_TR case displays stronger positive differences over the northeastern part of the domain (up to +0.25) which do not occur for the HCONV_TR experiment. Figure 2,*j* indicates that the Low sub-grid Convective Transport case (LCONV_TR) has lower absolute differences. A dipole of positive and negative differences (which means higher and lower AOD than the base case) is found over all the domain, a bit stronger over the fire-affected area.

Finally, turning off the scavenging (Figure 2,*l*; NO_WS experiment) leads to positive differences over a large part of the area with values higher than +0.2 over the north and west zones of the target domain. Moreover, temporal and spatial mean AOD difference (0.04) is the second largest even though there are not many clouds in the studied domain (see Figure 3 in Supplementary Material). This implies that wet scavenging could be really important when there are clouds present.

## 3.2 Optical properties and concentration profiles of different species: disentangling the causes of AOD changes

In order to disentangle the cause of the differences in the sensitivity tests, this section discusses the temporal mean of the vertical profiles of optical properties and concentration of several chemical species over specific locations of the target area. Figure 2, top-left displays the spot where the vertical profiles are estimated. The choice of these locations claims to bring light to the behaviour aloft over different places in the target area. Because of that, the locations where the temporal mean of AOD
was minimum and maximum, respectively, were selected and named as Min-AOD and Max-AOD. A profile over Moscow, one of the most fire-affected cities, was also chosen to evaluate the fire plume effect downwind.

In addition to $\alpha$, $\beta$ and lidar ratio (LR), concentrations for different species were evaluated: elemental carbon (EC), primary organic aerosol (POA), secondary organic aerosol (SOA), sea salt (SEA), nitrate ($NO_3^-$), ammonia ($NH_4^-$) and sulphate ($SO_4^{2-}$).

Vertical profiles over the Max-AOD location are shown in Figure 3. $\alpha$ and $\beta$ have similar shapes. The base case shows a
profile with high values (above $0.6\,\mathrm{km^{-1}}$ for $\alpha$ and below $0.02\,\mathrm{km^{-1}sr^{-1}}$ for $\beta$) at the surface. Both values decrease with height until around $0.2\,\mathrm{km^{-1}}$ for $\alpha$ and $0.005\,\mathrm{km^{-1}sr^{-1}}$ for $\beta$ at 900 hPa. Afterwards, values increase again to $0.3\,\mathrm{km^{-1}}$ for $\alpha$ and $0.01\,\mathrm{km^{-1}sr^{-1}}$ for $\beta$ at around 800 hPa (indicating the presence of aerosols associated to fire emissions aloft), where hereinafter decrease. Values are close to 0 above 600 hPa.

LR represents the ratio of the extinction and the backscatter coefficients and is usually used to characterize the type of
particles. This variable ranges from 1 to $100\,\mathrm{sr^{-1}}$ (Fernald et al., 1972). Following this definition, low LR values are expected for large and scattering particles and high LRs are expected for absorbing particles. Typical LRs at 532 nm are $20\text{-}35\,\mathrm{sr^{-1}}$ for sea salt, $40\text{-}70\,\mathrm{sr^{-1}}$ for desert dust, $70\text{-}100\,\mathrm{sr^{-1}}$ for biomass burning aerosols and $45\text{-}75\,\mathrm{sr}{-}1$ for urban/continental aerosols (Müller et al., 2007). The vertical profile of LR displays low values of around $35\,\mathrm{sr^{-1}}$ at low heights. LR increases to values between

50 and 60 $sr^{-1}$ around 700 hPa. Higher up, between 500 and 300 hPa, LR reaches values around 65 $sr^{-1}$ which again, above 300 hPa, decrease to 35 sr $^{-1}$. It is noticeable that LR values over the MIN-AOD location for most simulations (but NO_DD, where LR is close to 30-40 sr $^{-1}$) are similar to those values expected by the scientific literature (e.g Mielonen et al., 2013) for areas with biomass burning aerosols. However, it should be born in mind that MIN-AOD location is affected principally by sea salt, and therefore LR seems to be overestimated over the MIN-AOD location in most experiments. Moreover, extinction and backscatter modelling profiles shape are similar (rather constant at levels close to the surface), which is not found in most of the observed LR profiles. This could be ascribed to a model misrepresentation of extinction and backscatter modelling profiles. For example, Mielonen et al. (2013) measured the LR during the same forest-fire event in Finland. These authors found LR values of 60-70 $sr^{-1}$ for layers below 2 km, pointing to a mixture of biomass burning aerosols and other less absorbing aerosols. Conversely, in the upper layers the LRs were around 55 $sr^{-1}$, which indicated the presence of polluted dust. This reveals the misrepresentation in the LRs by our simulations, which estimate LRs around 35 sr$-1$ (typical LR values for sea salt particles) over areas with a high concentration of biomass burning aerosols (LR should typically reach values higher than 60 $sr^{-1}$).

In order to assess which species has the strongest influence on $\alpha$ and $\beta$, and also which chemical species presents the highest sensitivity in the designed experiments, profiles for the different species are shown in Figure 3, 5 and 7. Overall, total concentration is highly determined by the dry concentration, as expected for a heatwave episode. In addition, Figures 4, 6 and 8 quantify the mean absolute error (MAE) of each experiment with respect to the base case, and in colors, the normalized MAE (NMAE). MAE has been estimated by averaging the absolute error of each experiment regarding the base case at each model level. NMAE is the absolute error divided by the base case at each level and then averaged along the column. The NMAE analysis illustrates the relative change of each specie and optical properties and helps to the intercomparison between the sensitivity test.

### 3.2.1 Sensitivity to the relative humidity

When the sensitivity tests are evaluated over the MAX-AOD location, the experiments changing the RH present a singular response. When RH increases (H05RH and H1RH), the profile of optical properties also increases, as well as the AOD. MAE for the profiles (Figure 4) of $\alpha$ ($\beta$) are 0.0101 (0.0005) and 0.0159 (0.0004), for the case in which RH is scaled to 0.5 % (H05RH) and 1 % (H1RH) respectively, and NMAE are 0.4 (0.4) and 0.6 (0.5). These differences could be caused by the high dependence of AOD on water uptake, which finally depends on RH, as indicated by Ginoux et al. (2006); Yoon and Kim (2006); Altaratz et al. (2013); Palacios-Peña et al. (2017, 2018, 2019a). Thus, an increase in RH affects the hygroscopic growth, resulting in larger particles. For this reason, a reduction of optical properties is expected when RH decreases (L10RH experiment). However, the results indicate an increase in AOD and profiles of extinction and backscatter coefficients (MAE 0.0162 and NMAE 0.6 for $\alpha$; and 0.0005 and 0.7 for $\beta$). This response is the result of an increase of $NO_3^-$ (MAE 0.8209 and NMAE 0.6) and, in particular, of SOA (MAE 0.2054 and NMAE 0.9).

The concentrations of inorganic species are controlled by the so-called sulphate-ammonium-nitrate-water equilibrium (Seinfeld and Pandis, 2006). $NO_3^-$ and $NH_4^-$ present a deliquescence RH of approximately 60 % (Saxena et al., 1986). However, $SO_4^{2-}$ absorbs water at nearly all RH. As exposed by Weigum et al. (2016), due to the RH absorption by the $SO_4^{2-}$, the

equilibrium is dominated by the reaction in which ammonia neutralizes sulphuric acid and drives the equilibrium towards the aerosol phase ($(NH_4)_2SO_4$). Therefore, ammonia can neutralize nitrate resulting in aerosol phase ($NH_4NO_3$) only when the total amount of sulphate has been neutralized (i.e. in areas with high concentrations of ammonia and/or low concentrations of sulphate). At this point, sulphate concentrations remain constant, and nitrate increases with aerosol water content.

This sulphate-ammonium-nitrate-water equilibrium explains the behaviour of the inorganic species. For the highest RH case (H1RH), $NO_3^-$ concentration shows a considerable increase while $SO_4^{2-}$ slightly increases. This could be influenced by an increase of the RH favouring the $NO_3^-$ formation together with a high sulphate concentration for which most of the sulphate has been neutralized.

However, in the case with a reduction of the RH in a 10 % (L10RH), $NO_3^-$ displays a similar concentration as the base case at surface levels and around 800 hPa. Throughout the rest of the profile concentration is higher than in the base case but not as higher as in the H1RH case. Meanwhile, $SO_4^{2-}$ concentrations are much higher than for the base case. Sulphate concentrations are favoured by its low deliquescence point which promotes its formation. In spite of that, at higher levels, sulphate concentrations were at the point in which most of the sulphate has been neutralized favouring $NO_3^-$ formation, producing higher $NO_3^-$ concentrations in the L10RH case.

The H05RH (RH scaled to 1.005) experiment shows optical properties and concentration profiles closer to the base case, which can be caused by the low RH modification, so that inorganic species are not highly affected by this change.

Changes in the profiles of inorganic species do not clarify the results found for the modifications in the profiles of optical properties (and AOD). These modifications are led by changes in SOA. In both H1RH (RH scaled to 1.1) and L10RH (RH scaled to 0.9), SOA profiles depict an increase in their concentrations resulting in an increase of $\alpha$ and hence AOD. Moreover, this increase is higher for the L10RH case. This positive variation in SOA profiles are explained by the use of the VBS mechanism (Ahmadov et al., 2012). As pointed out by Tuccella et al. (2015), in this mechanism volatile organic compounds(VOC) are oxidized by reactions with the hydroxyl radical (OH), $O_3$, and nitrate radical ($NO_3^-$, producing organic mass in two different regimes of high and low $NO_x$). In the former, organic peroxy radicals react with nitrogen monoxide (NO); conversely, in the latter organic peroxy radicals react with other organic peroxy radicals. The organic matter produced is partitioned into aerosol and gas phase assuming a pseudo-ideal partition.

Thus, SOA profiles for the RH case depict two different types of behaviour: (1) Above 950 hPa (around the PBL height, see Figure 1 in the Supplementary Material) the shape of the $NO_x$ and SOA profiles are similar, and thus, at these vertical levels, variations in SOA concentrations may be due to an increase in $NO_x$ concentrations at low-$NO_x$ conditions (less than 30 ppb or around 55 $\mu$g m$^{-3}$; Sarrafzadeh et al., 2016); (2) Below 950 hPa the RH effect is added to the effect of $NO_x$ described above in (1). Therefore, in the H1RH case, SOA are higher because the concentration of this species increases due to $NO_x$ oxidation and RH, meanwhile in the L10RH case the positive variation of the concentration of SOA caused by the RH is limited. That means that this variation depends more of RH modifications (see Figure 2 in the Supplementary Material) than $NO_x$ oxidation.

Over the MIN-AOD, the RH scaled to 0.9 (L10RH; NMAE > 0.6 except for SEA, 0.1) should be highlighted. Despite L10RH does not provoke a strong difference in AOD, changes in organic species are relatively strong and are similar to those

changes in $\beta$ profile. A reduction of RH may favour the increase of the concentration of these species. $\alpha$ profile is similar to $NO_3^-$. In this case, these changes could be due to the actions of the nitrate-ammonia-sulphate equilibrium.

Finally, to elucidate the response of the different experiments over a downwind location, profiles over Moscow are shown in Figure 7. The response for most of the experiments is similar as over the MAX-AOD location; but in this case L10RH (RH scaled to 0.9) experiment shows a stronger response (NMAE > 1.5 for most of the variables) due to higher $NO_3^-$ concentrations. Over this location RH is higher than over the MAX-AOD, favouring the formation of $NO_3^-$. POA displays higher concentrations for the L10RH case, likely due to a competition of SOA formation between $NO_3^-$ and POA.

### 3.2.2 Sensitivity to dry deposition

Regarding dry deposition over the MAX-AOD location, the no dry deposition case (NO_DD) shows an increase in the AOD over the target area and displays higher $\alpha$ and $\beta$ values than for the base case at near-surface levels. However, above 950 hPa (around the PBL height, see Figure 1 in the Supplementary Material), the optical profiles decrease to levels lower than those for the base case. Despite this decrease aloft, total AOD increases (Figure 2) likely because the highest concentrations for chemical species are located at these levels. With respect to the different species, all of them present higher concentrations than the base case, in particular at levels below 950 hPa. MAE (NMAE) of $\alpha$ and $\beta$ for this experiment are 0.0283 (1.1) and 0.0008 (1.1, Figure 4).

Changes in dry deposition experiments occur in those modes where modifications were implemented (Figure 4 in the Supplementary Material). When modifying the deposition of the accumulation mode, the Aiken mode does not present important changes and thus the observed variations come from the accumulation mode. However, when modifications are implemented in the deposition of the Aitken mode, both modes are affected, since particles in the Aitken mode quickly experience coagulation processes and turn into particles in the accumulation mode.

A higher AOD is also found for the LDDV_AIT case (low dry deposition velocity in the Aitken mode). For this experiment, $\alpha$ (MAE 0.0205 and NMAE 0.8) and $\beta$ (MAE 0.0005 and NMAE 0.7) exhibit higher values at the surface (around 1000 hPa) and between 900 and above 600 hPa. With respect to the profile of the different species, those emitted directly into the atmosphere (primary species) present higher concentrations than the base case at surface levels (around 1000 hPa and below 800 hPa, respectively). This is observed for POA (MAE 2.1988 and NMAE 0.7) and SEA (MAE 0.0154 and NMAE 0.4). However, secondary aerosol; which are not directly emitted and are products of atmospheric chemistry; such as SOA (NMAE > 0.8 and MAE 0.2283) and most of the secondary inorganic species have their concentrations peak at a higher altitude than those in the base case between 900 and 600 hPa. These two facts explain the response of the profiles for the optical properties.

As expected, both high DDV experiments (HDDV_AIT and HDDV_ACC, in the Aitken and the accumulation mode respectively) exhibit a reduction of AOD, in particular over the fire area. The response of the profiles of optical properties is similar for both cases and for most of the species. For example, MAE (NMAE) are 0.0365 (0.8) and 0.0392 (1.5) for $\alpha$. Only SEA shows a different behaviour between the increase of DDV for Aitken (NMAE 0.7) or accumulation mode (NMAE 0.8). The reduction of the total concentration of SEA is higher when DDV is modified in the accumulation mode. This is produced because this species presents most of its concentrations in the Greenfield gap (particle radii of the range of 0.1–1 $\mu$m where

Brownian motion is not large anymore and gravitational settling is not yet important; Greenfield, 1957; Ladino et al., 2011), the accumulation and the coarse mode and not in Aitken. Regarding organic species (EC, POA and SOA), concentrations are a bit lower when the DDV is modified in the accumulation mode, probably because most of the mass of these species is in this mode (NMAE around 1.4 for all of them). This response is similar to those experiments for $SO_4^{2-}$, but it is the contrary for

$NO_3^-$ because of the the action of the nitrate-sulphate-ammonium equilibrium.

The low dry deposition velocity in the accumulation mode (LDDV_ACC) experiment does not show the *a priori* expected response. AOD decreases over the fires; also optical properties profiles displays lower values: MAE (NMAE) 0.0331(1.3) for $\alpha$ and 0.001(1.3) for $\beta$. When the profiles are analyzed, the response differs between species. EC, POA and $NO_3^-$ shows a slight reduction in their concentration, and SOA exhibits a large reduction. Conversely, $SO_4^{2-}$ and SEA display higher concentrations,

in particular, at near-surface levels. The response of these latter is the expected when DDV is decreased in the accumulation mode but, despite this increase, the decrease of AOD is the result of the large reduction of SOA concentrations (NMAE 1.1). These SOA reductions may occur due to the increase in $SO_4^{2-}$ concentrations (NMAE 1). By modifying the DDV, $SO_4^{2-}$ concentrations increase, then the nitrate-sulphate-ammonium equilibrium results in a reduction of $NO_3^-$, which influences SOA formation (as explained above) by decreasing their concentration.

Due to the different behaviour over the MIN-AOD location with respect to those areas affected by wildfires, the no dry deposition (NO_DD; NMAE > 0.9 for all the variables) should be highlighted. For NO_DD, $\beta$ profile is similar to the profiles of organic species (EC, POA and SOA) as well as $NH^{-4}$ and $S_4^{-2}$ while $\alpha$ is similar to $NO_3^-$. Organic species present a higher concentration when dry deposition is turned off, resulting in an increase of $\beta$. However, $NO_3^-$ decreases, probably due to its competition with $SO_4^{2-}$ (which increases), leading to a decrease close to the surface of $\alpha$. However, these changes in optical

properties profiles are not highly represented by a strong modification of total AOD.

Over the Moscow location, the NO_DD experiment also displays a strong response (NMAE > 1 for all the variables). This response is explained by an increase in the concentration of all the species, in particular, at the surface due to the effect of turning off dry deposition resulting in an increase of $\alpha$ and $\beta$.

### 3.2.3 Sensitivity to sub-grid convective transport

When sub-grid convection is modified, in both experiments NO_CONV_TR (convective transport turned off) and HCONV_TR (scaled by 1.5) there is an AOD reduction over the fire area. This decrease is also reflected in optical properties over the MAX-AOD location and for most of the species (NMAE > 0.8 in both experiments) except $SO_4^{2-}$. For POA, EC and $SO_4^{2-}$, the NO_CONV_TR experiment exhibits a concentration profile similar to the base case with slightly higher concentrations at surface levels and lower at higher levels (NMAE 0.4). The opposite behaviour is displayed by SEA concentrations. Moreover,

SOA, $NO^{-3}$ and $NH^{-4}$ concentration are constantly smaller than the base case. However, the $SO_4^{2-}$ concentration profile for the HCONV_TR experiment shows lower concentrations (NMAE 1.2). Both responses could be caused by modifications in sub-grid convective transport. When this transport is turned off there is a decrease in the particle mixing in the atmosphere and small differences with the base case are found. However, when this transport is increased, involving an increase in all-direction

convective transport and not only updraft convection, there is a higher mixing of particles. This fact can favour the transport to
levels closer to the surface and then enhance the deposition processes.

For the HCONV_TR experiment, the behaviour of $SO_4^{2-}$ is similar to the rest of the species. Thus, the modification in sub-grid convective transport controls the response of this experiment. However, for the NO_CONV_TR test, the rest of the species behave differently than $SO_4^{2-}$. $NO_3^-$ strongly decreases due to the effect of the nitrate-ammonium-sulphate equilibrium in which the sulphate is an obstacle for $NO_3^-$ formation. This low $NO_3^-$ concentration results in a decrease of the SOA formation
and consequently its concentration. This finally leads to a decrease of $\alpha$ and AOD. The response of LCONV_TR (convective transport scaled to 0.5) shows a transition between the two extreme cases (NMAE around 1 for all of the variables except SEA, 0.1, and $SO_4^{2-}$, 0.4).

Over the MIN-AOD location, the behaviour observed for the LCONV_TR experiment (convective transport scaled to 0.5) is also noteworthy albeit NMAE does not have a strong response. AOD is not strongly modified but the profiles of optical
properties how a peak around the PBL height. This peak is due to an increase in the concentrations of EC, POA, SOA and $NO_3^-$. For the organic species, this increase can be due to the modification in the sub-grid convective transport. The presence of these species at this level seems to favour the formation of $NO_3^-$ instead of $SO_4^{2-}$.

### 3.2.4   Sensitivity to wet scavenging

The modification of wet scavenging over the MAX-AOD location displays a slight reduction of AOD, which is the result of
lower $\alpha$ and reduced concentration of species above the PBL (at 950 hPa). NMAE is < 0.8 for most of the studied variables. This reduction is observed despite the inorganic species (SEA, $NO_3^-$, $NH_4^-$ and $SO_4^{2-}$) show higher concentrations at the lowest levels. SOA also displays a higher concentration below 800 hPa but with smaller changes than for inorganic species. This highlights the high impact of organic species on optical properties. All the observed changes can be attributed to changes in the aqueous phase reactions because over these locations stratiform clouds were not present.
To conduct the analysis where clouds were formed during the 2010 wildfires episode (see Figure 3 in the Supplementary Material), the MIN-AOD location is shown in Figure 5. Over this location, the NO_WS experiment has the strongest response regarding optical properties profiles and concentrations for different species. NMAE is above 1.5 for all the studied variables. The profiles of optical properties depict much higher values than for the base case, which are also observed in all of the species. This could be due to the fact that over this area stratiform clouds were present, so the effect of wet scavenging is important
over this location.

It should also be highlighted, the profile shape of EC and POA over the MIN-AOD and Moscow show larger differences than over the MAX-AOD area for the different experiments. These differences in the shapes of the profiles could be attributed to species which are not directly emitted over the MIN-AOD and Moscow areas thus, the vertical distribution could be influenced by transport processes. Moreover, the farther the location is, the more different the shape of the vertical profile is.

 **4 Discussion**

The main finding of this work is the non-linear response exhibited by AOD when characterizing its sensitivity to different key processes. This response is highly dependent on the thermodynamics equilibrium sulphate-nitrate-SOA, in which also water and ammonia play an important role. Moreover, and probably due to the nature of this episode (heatwave/wildfires), SOA shows a high impact on the representation of aerosol optical properties, as also found by Regayre et al. (2018) and Yoshioka et al. (2019). These works highlighted a large uncertainty in effective radiative forcing related to ARI because of the presence of carbonaceous aerosols in high-emission months and in regions close to emission sources. However, under other conditions, the global influence of anthropogenic sulphate aerosol presented a significant influence on AOD estimations (not only due to emission but also to transport or lifetime; Kasoar et al., 2016; Regayre et al., 2018; Yoshioka et al., 2019). This behaviour was also observed, in a lesser extent, for nitrate (Balzarini et al., 2015). Thus, a large effort should be devoted to the process-understanding of this non-linear response from several key sources (RH, convective transport, dry deposition and other aerosol processes) and the improvement of the representation of the equilibrium sulphate-nitrate-ammonia-water in models for reducing the uncertainty related to aerosols in on-line coupled models.

From a global point of view, different works identified the processes evaluated in this work as important sources of uncertainty when characterizing aerosol optical properties and/or radiative forcing (which is highly influenced by the former). Regayre et al. (2018) found the deposition rate of aerosols and aerosol precursors (gases) to be the most important causes of the uncertainty related to effective radiative forcing. Also, dry deposition was the most important process for global mean CCN uncertainty (Lee et al., 2013), a source of uncertainty in AOD representation (Romakkaniemi et al., 2012). Although this process presents large uncertainties when estimating AOD, its importance is limited over a fire-affected region. Thus, attention should be paid in the evaluation of the uncertainty of this process depending on the scale, since the impacts of this process would likely be stronger over other regions, making it important globally. As pointed out by Regayre et al. (2018), some causes of uncertainty in radiative forcing could be because they cause at least a small amount of uncertainty in nearly all regions or because they are the largest causes in some regions. Both Lee et al. (2013) and Regayre et al. (2018) used global models (GLOMAP-mode within the TOMCAT global 3-D offline chemistry transport model and HadGEM-UKCA model, respectively) during a whole year. Moreover, it should be highlighted that CCN uncertainty affects not only AOD representation but also to radiative forcing uncertainties due to ACI (Lee et al., 2013).

Similar results were found by Kipling et al. (2016) for convective transport using the HadGEM3-UKCA model. This process was found to be very important when controlling the vertical profile of all aerosol components by mass. In addition, previous works as Palacios-Peña et al. (2018) and Palacios-Peña et al. (2019a) found that a misrepresentation of aerosol vertical profile could lead to uncertainties in the representation of AOD. On the other hand, Croft et al. (2012) evaluated the uncertainty due to different assumptions for the wet scavenging of aerosol and found a 20 to 35 % uncertainty in simulated global, annual mean of AOD using the ECAM5-HAM model. However, the findings in our work regarding wet scavenging were lower due to the type of episode selected (without extensive clouds).

Another source of uncertainty is that related with general circulation. In this sense, Nordling et al. (2019) demonstrated a significant uncertainty in regional climate responses due to differences in circulation even with perfect aerosol descriptions.

In addition, Brunner et al. (2015) pointed out the need for improving the simulations of meteorological parameters relevant for air quality. On the other hand, other works found an effect on meteorological variables, and thus, in circulation responses when aerosol effects are taken into account. This source of error is more relevant during the summer and near large sources of pollution (Makar et al., 2015b; Baró et al., 2017), conditions that are similar to the episode analyzed here. These works show an impact on shortwave downwelling radiation at the surface, temperature, RH and PBL height due to the inclusion of

aerosol effects which again could affect AOD uncertainties. Moreover, Kong et al. (2015) evidenced an improvement in the skill of meteorological variables when aerosol radiation effects were included. Thus, the uncertainties in the representation of the vertical distribution of aerosols and their optical properties revealed in this work could be limited to the influence of the regional circulation response, which in turn could again impact the aerosol distribution. Henceforth, a reduction in this aerosol uncertainty could at the same time reduce the uncertainty in the response of the circulation and thus, the evaluation of

uncertainty could be constrained only to uncertainties in circulations mechanisms, as pointed out by Nordling et al. (2019).

Other important sources of uncertainty in the representation of aerosol optical properties among those evaluated here are the aerosol emissions (Granier et al., 2011; Soares et al., 2015), representations of complex sub-grid processes (Weigum et al., 2016), aerosol processes (Croft et al., 2012), subsequent feedbacks on atmospheric dynamics (Booth et al., 2012; Bollasina et al., 2013; Villarini and Vecchi, 2013; Makar et al., 2015b; Baró et al., 2017; Nordling et al., 2019; Palacios-Peña et al.,

2019b), aerosol mixing (Zhang et al., 2012; Curci et al., 2019) and aerosol size distribution (Tegen and Lacis, 1996; Claquin et al., 1998; Eck et al., 1999; Haywood and Boucher, 2000; Romakkaniemi et al., 2012; Obiso et al., 2017; Obiso and Jorba, 2018; Palacios-Peña et al., 2020). Another source of uncertainty is the choice of the aerosol-chemical mechanisms which was pointed out by Balzarini et al. (2015) and Palacios-Peña et al. (2018, 2019a).

In order to simplify the approach, this study has been conducted using only one model; however, differences among models

and how these represent the life cycle of aerosols should be kept in mind (Randles et al., 2013; Kim et al., 2014; Mann et al., 2014; Tsigaridis et al., 2014; Lacagnina et al., 2015; Pan et al., 2015; Ghan et al., 2016; Kipling et al., 2016; Koffi et al., 2016; Palacios-Peña et al., 2018, 2019a; Nordling et al., 2019) and similar studies with other modelling approaches are necessary for an overall knowledge of these uncertainties.

## 5 Summary and Conclusions

Aerosol optical properties (e.g. AOD) are highly influenced by the vertical distribution of atmospheric aerosols, which also condition the representation of ARI and ACI processes and their uncertainty. Thus, a key issue in climate modelling is the assessment of the uncertainty in the representation of aerosol optical properties. This work assesses the sensitivity of aerosol optical properties and the aerosol vertical distribution to several key physical processes. To achieve this objective, sensitivity runs modifying RH, dry deposition, sub-grid convective transport and wet scavenging have been carried out for the 2010

Russian heatwave/wildfires episode with the WRF-Chem regional fully coupled model. The findings in this work could help

improving modelling strategies for aerosol representation, giving some initial guidelines about what parameters could be mis-represented or are the most sensitive to the vertical mixing.

Results indicate that there is a non-linear response of AOD to different key processes. For example, both an increase and a decrease in the RH results in higher AOD values. A similar non-linear response is found when reducing the dry deposition velocity; in particular, for the accumulation mode, where the concentration of several species increases (a decrease might be *a priori* expected). Also the modifications in the sub-grid convective transport exhibit a non-linear response because both the increase and offset of this process leads to a reduction in the AOD over the fire area. Similar non-linear responses were previously found, among others, by Lee et al. (2013); Kipling et al. (2016) using different models and experiments; and by Weigum et al. (2016) using the WRF-Chem model (as also done in this contribution).

With respect to the quantification of the sensitivity, modifying RH by a factor of 0.9 leads to the highest AOD differences (0.6). This high sensitivity is followed in relevance by scaling vertical convective transport (with AOD differences around -0.4) and dry deposition (AOD differences up to -0.35 and 0.3).

However, when RH increases (1.005 or 1.01 scaling factors), the response is weaker (AOD differences lower than 0.15) than when RH decreases. This is because the scaling to high RH values is smaller since an important supersaturation (above 1-2 %; Devenish et al., 2016) is not realistic in climate models. When the RH slightly increases, AOD changes are conditioned by the water uptake by particles and hence the humidity contributes to the modification of the size of particles by hygroscopic growth (see H05RH experiment). In this case, no large changes in concentrations are found. Nevertheless, for larger increases in RH (H1RH), changes in AOD are dominated by changes in nitrate and SOA. These changes in SOA are controlled by two mechanisms of particles formation: (1) the first mechanism, the nitrate-ammonia-sulphate equilibrium, explains the changes found for $SO_4^{2-}$ and $NO_3^-$. Summarizing, the amount of sulphate domains this equilibrium in which ammonia can neutralize nitrate only when there is a high concentration of ammonia and/or low concentrations of sulphate. Henceforth, if most of the $SO_4^{2-}$ concentration has been neutralized, an increase in RH favours $NO_3^-$ formation. Moreover, in low RH conditions, $NO_3^-$ formation is possible only under low $SO_4^{2-}$ concentrations; (2) the second mechanism which controls SOA formation is the implemented VBS mechanism (Ahmadov et al., 2012; Tuccella et al., 2015). In our experiments, VOC are oxidized by reactions with nitrate radical in the regime of low $NO_x$ and SOA increases as $NO_3^-$ concentrations, as described by Sarrafzadeh et al. (2016).

Dry deposition presents a higher impact for the accumulation mode (NMAE higher than 1.4) than for the Aitken mode (NMAE around 1.3) because a higher mass of fire particles is emitted into this mode. Over the MAX-AOD location, switching off the dry deposition does not have a strong impact on AOD, but it does over the rest of the domain. Over MAX-AOD location, particles are directly emitted into the atmosphere, while over other locations transport phenomena govern the concentrations. In general, when dry deposition is suppressed or reduced, AOD increases and conversely when it is increased, AOD decreases. However, the response over the MAX-AOD location of the decrease of dry deposition for the accumulation mode is noticeable because a decrease in the dry deposition in this mode significantly increases $SO_4^{2-}$ concentrations. Thus, the nitrate-ammonia-sulphate equilibrium reduces $NO_3^-$ leading to a reduction of SOA and then AOD.

The suppression and the increase of the vertical convective transport also presents an impact on the aerosol vertical distribution. When the vertical convective transport is increased all the species show a similar response. This modification implies an increase of the transport not only upwards but also in all directions, increasing the mixing of particles which can favour the transport from upper layers to the surface, hence enhancing deposition. However, when the sub-grid convective transport is suppressed the nitrate-ammonia-sulphate equilibrium and the SOA formation mechanisms play an important role. A reduction

in the vertical convective transport, which can reduce the mixing of particles, results in significant changes of AOD but over regions away from the sources (main emission areas), in particular, over the MIN-AOD spot.

       Wet scavenging does not significantly impact the vertical aerosol mass due to the type of episode selected as case study (heatwave with clear skies). There is an impact over the MIN-AOD location because this is a cloudy area during the period of the episode.

Regarding the LR, simulated values of this variable are remarkably different from those observed in the scientific literature, mainly over fire affected areas. In those areas where high LR are expected due to the presence of biomass burning particles, simulations estimate lower LR (and viceversa). It should be also pointed out that most of the species show relatively larger differences when they are considered far away from the emission areas. Thus, as pointed out by Lee et al. (2013), the uncertainty in aerosol microphysical processes becomes increasingly important in remote regions (far from the source of emissions).

To summarize, the sulphate-nitrate-SOA formation is the process with the largest sensitivity and hence the process whose uncertainty can have a larger impact on AOD representation. Changes in this process could come mainly from modifications in RH, dry deposition or vertical convective transport. Alone, dry deposition also presents a high sensitivity which influences AOD representation.

       Last, it should be noticed that the processes evaluated here are not the only processes that might condition the uncertainty

in aerosol properties. The selection of these experiments has been based on their relevance according to the available literature and their experimental design has been constrained by the high computational cost of these on-line coupled chemistry-meteorological simulations. In this sense, further studies addressing the reduction of the demonstrated uncertainties are needed. Reducing uncertainties of AOD and aerosol representation implies the reduction of uncertainties in the representation of aerosol effects, both ARI (by AOD) and ACI (by improvement in microphysical properties) providing more reliable weather predictions

and climatic simulations.

*Data availability.* The data is available upon contacting the corresponding author (pedro.jimenezguerrero@um.es)

*Author contributions.* LP-P wrote the manuscript, with contributions from PJ-G. LP-P and PS designed the experiments; LP-P conducted the numerical simulations and compiled all the experiments, with the support of RL-P. LP-P did the analysis, with the support of PS, RL-P and PJ-G.

*Competing interests.* The authors declare no conflict of interest.

*Acknowledgements.* The authors acknowledge the ACEX-CGL2017-87921-R project, funded by the Spanish Ministry of the Economy and Competitiveness and the European Regional Development Fund (ERDF/FEDER). L. P.-P. thanks the FPU14/05505 scholarship from the Spanish Ministry of Education, Culture and Sports and the ERASMUS+ program. P.S. acknowledges funding from the European Research Council (ERC) project constRaining the EffeCts of Aerosols on Precipitation (RECAP) under the European Union's Horizon 2020 research and innovation programme with grant agreement 724602.

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

**Table 1.** WRF-Chem physical and chemical configuration used in the sensitivity tests.

| Scheme | Option | Reference |
|---|---|---|
| **Physic** | | |
| Microphysics | Morrison | Morrison et al. (2009) |
| SW & LW radiation | RRTM | Iacono et al. (2008) |
| Planetary boundary layer | YSU | Hong et al. (2006) |
| Cumulus | Grell-Freitas | Grell and Freitas (2014) |
| Soil | Noah | Tewari et al. (2004) |
| **Chemistry** | | |
| Gas-phase | RACM-KPP | Stockwell et al. (1997) Geiger et al. (2003) |
| Aerosol | MADE/VBS | Ackermann et al. (1998) Tuccella et al. (2015) |
| Photolysis | Fast-J | Fast et al. (2006) |
| Dry Deposition | | Wesely (1989) |
| Wet Deposition | grid-scale | |
| ARI & ACI | ON | |

**Table 2.** Description of the experiments carried out to perform the sensitivity tests of aerosol to different processes; changes of relative humidity (RH), dry deposition (DDV), convective transport and wet scavenging.

| Experiment | Description |
|---|---|
| Base Case | – |
| L10RH | RH scaled to 0.9 in the aerosol module |
| H05RH | RH scaled to 1.005 in the aerosol module |
| H1RH | RH scaled to 1.01 in the aerosol module |
| NO_DD | No aerosol dry deposition (DD) |
| LDDV_AIT | DDV scaled to 0.5 for Aitken Mode |
| HDDV_AIT | DDV scaled to 2 for Aitken Mode |
| LDDV_ACC | DDV scaled to 0.1 for the Accumulation Mode |
| HDDV_ACC | DDV scaled to 10 for the Accumulation Mode |
| NO_CONV_TR | No sub-grid convective transport |
| LCONV_TR | Sub-grid convective transport scaled to 0.5 |
| HCONV_TR | Sub-grid convective transport scaled to 1.5 |
| NO_WS | No stratiform wet scavenging |

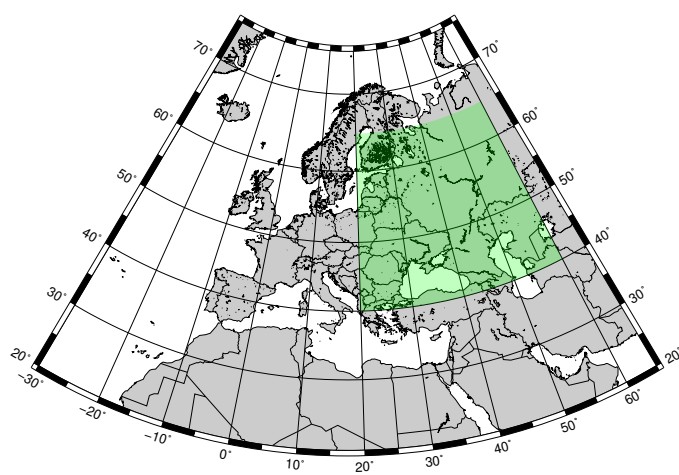

**Figure 1.** Simulated domain (grey) and fire-affected target area (green box).

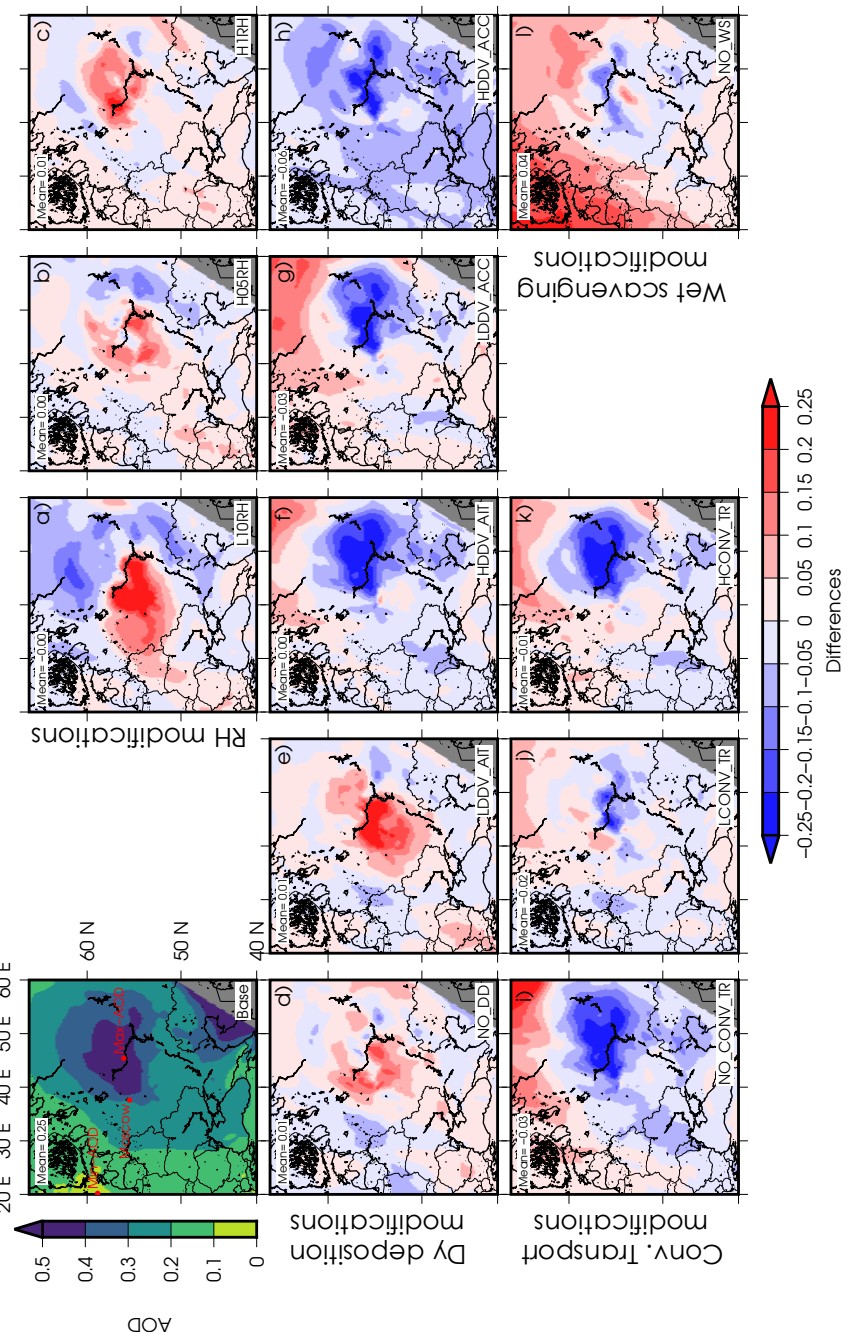

**Figure 2.** Modelled AOD at 550 nm for the base case (top-left) and mean bias differences between experiments and the base case. RH modifications at the top-right: a) scaled to 0.9 (L10RH); b) scaled to 1.005 (H05RH); and c) scaled to 1.01 (H1RH). Dry deposition modifications at the second row: d) the suppression (NO_DD); e) the low DDV for the Aitken mode (LDDV_AIT); f) the high (HDDV_AIT); g) the low DDV for the Accumulation mode (LDDV_ACC); and h) the high (HDDV_ACC). Sub-grid convective transport are in bottom-right row: i) the suppression (NO_CONV_TRANS); j) scaled to 0.5 (LCONV_TRANS) and k) scaled to 1.5 (HCONV_TRANS). Bottom-left panel, l), is the suppression of the wet scavenging.

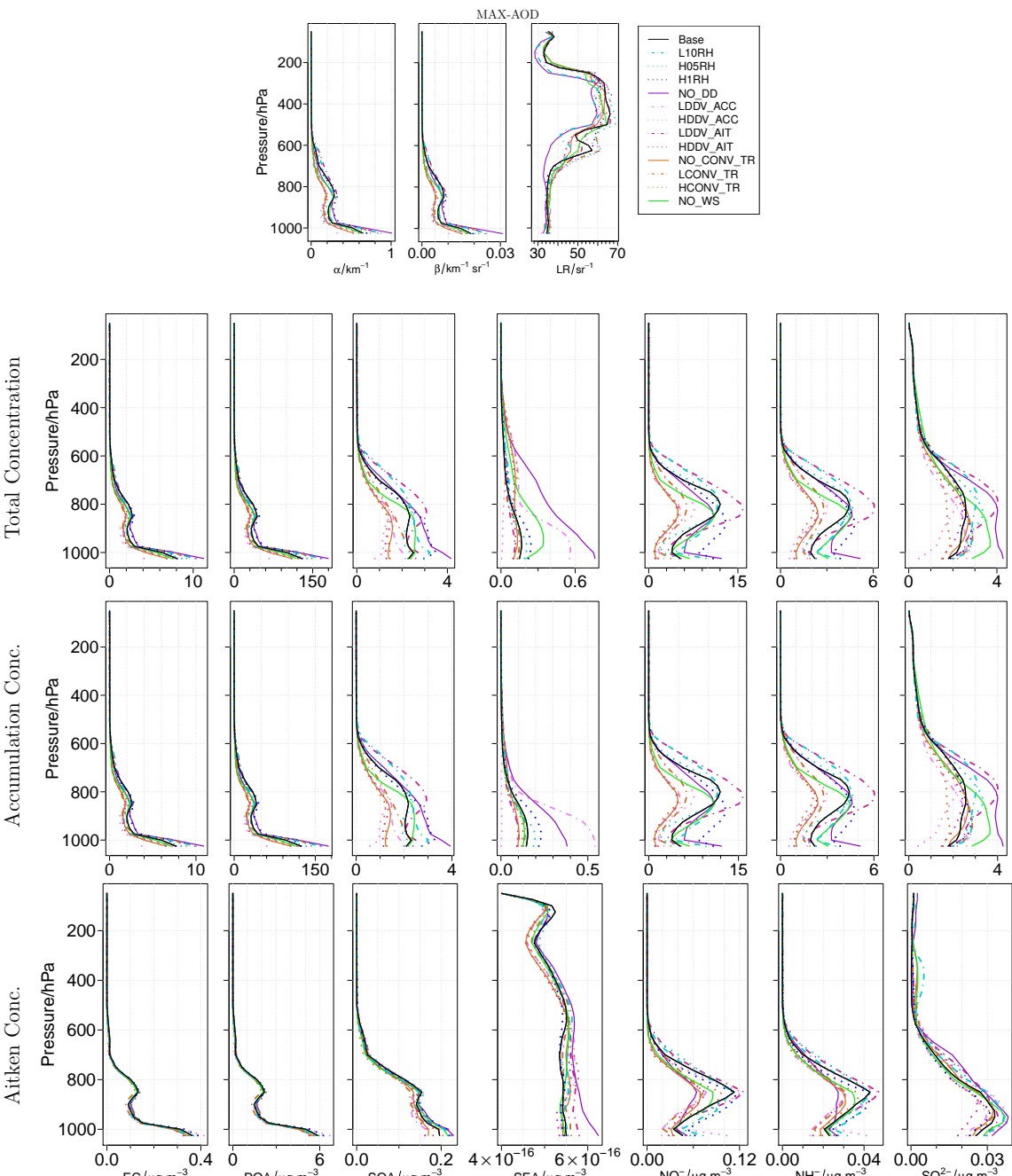

**Figure 3.** Profiles over the Max-AOD location. Top row shows the $\alpha$ (left), $\beta$ (centre) & LR (right). From second to bottom rows, columns display concentration of EC, POA, SOA, SEA, $NO_3^-$, $NH_4^-$ & $SO_4^{2-}$. The second row is for total concentration; the third for dry; and the bottom for wet. The solid black line represents the base case. The blue color is for RH sensitivity: the dotted dark is the high in 1 % (H1RH); the dotted light, the high in 0.5 % (H05RH); and the dotted-dashed light, the low in 10 % (L10RH). The violet color is for dry deposition. The solid dark is the no dry deposition (NO_DD). The rest dark are for the modification of DDV in the Aitken mode: the dotted is the high (HDDV_AIT); and the dotted-dashed, the low (LDDV_AIT). Similar but in light violet is for the accumulation mode: the dotted is the high (HDDV_ACC); and the dotted-dashed, the low (DDV_ACC). The brown color is for sub-grid convective transport: the solid, without it (NO_CONV_TR); the dotted, the high case (HCONC_TR); and the dotted-dashed, the low (LCONV_TR). The solid green represents the wet scavenging turned off (NO_WS).

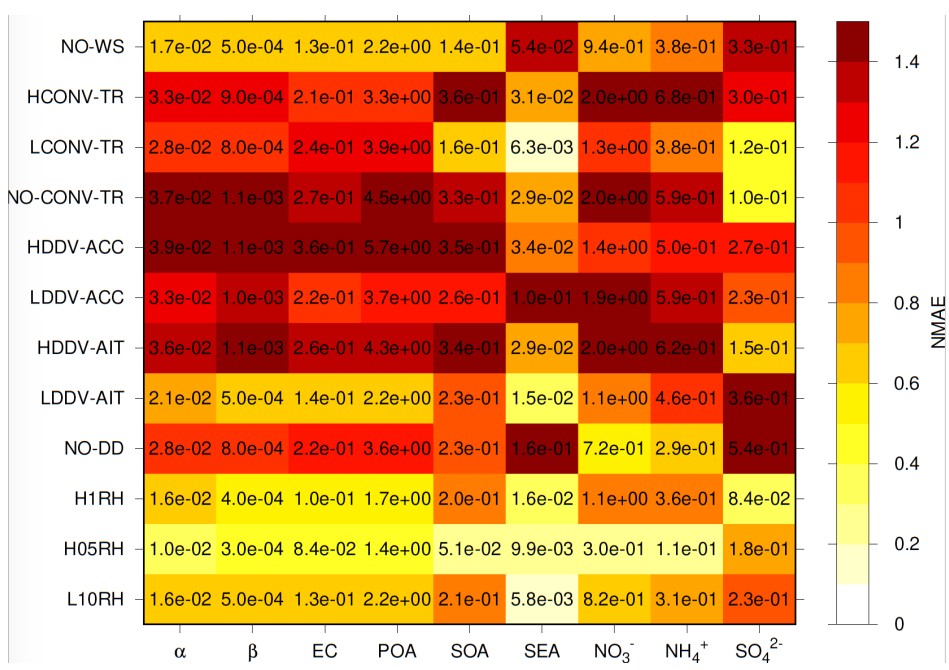

**Figure 4.** Normalized absolute differences (color) and absolute differences (numbers) between each experiment and the base case over the MAX-AOD location. Columns represent each variable and rows each experiment.

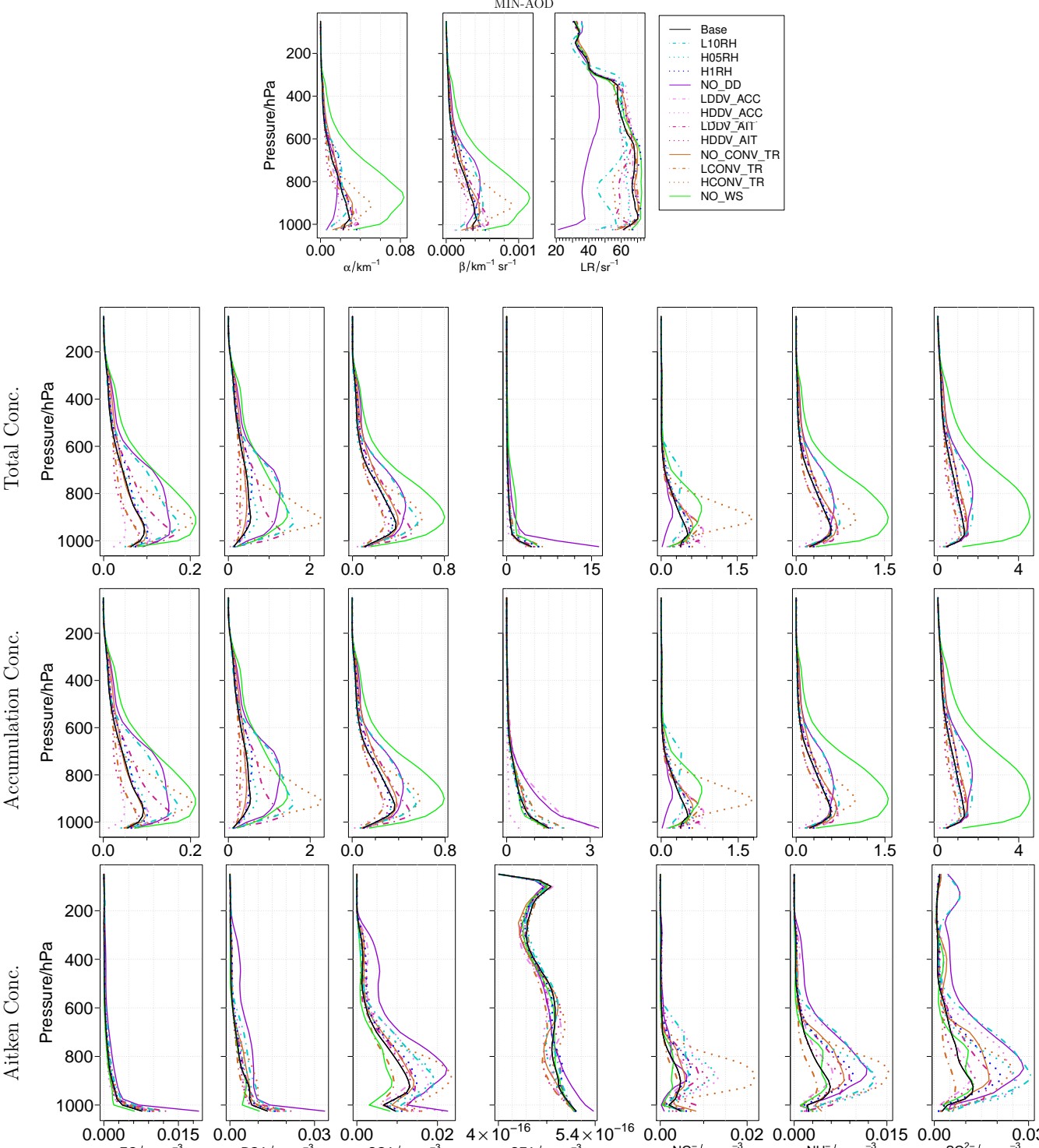

**Figure 5.** As Figure 3 but over the MIN-AOD location.

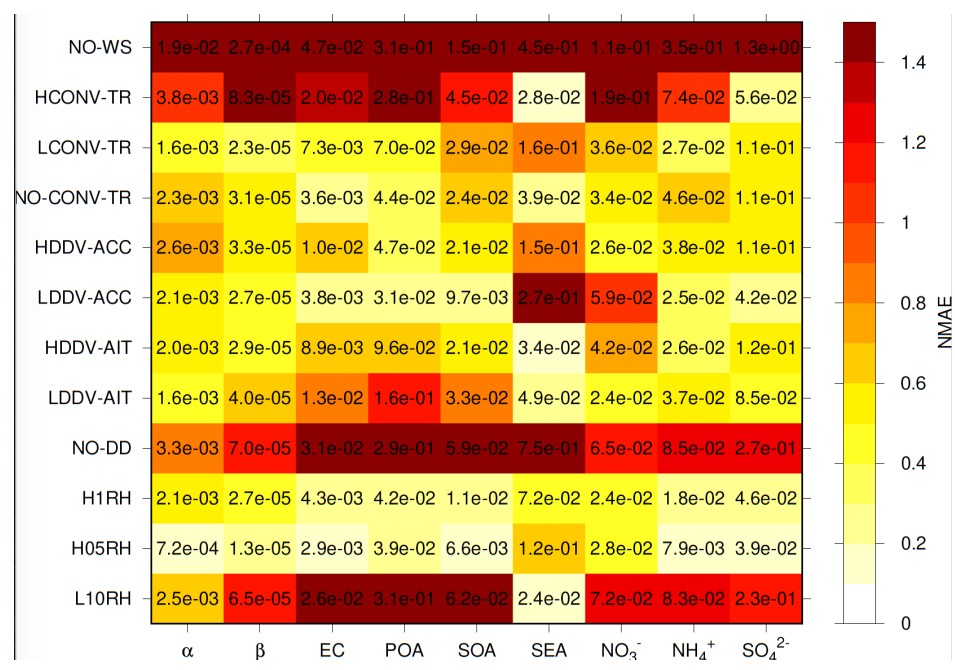

**Figure 6.** As Figure 4 but over MIN-AOD location.

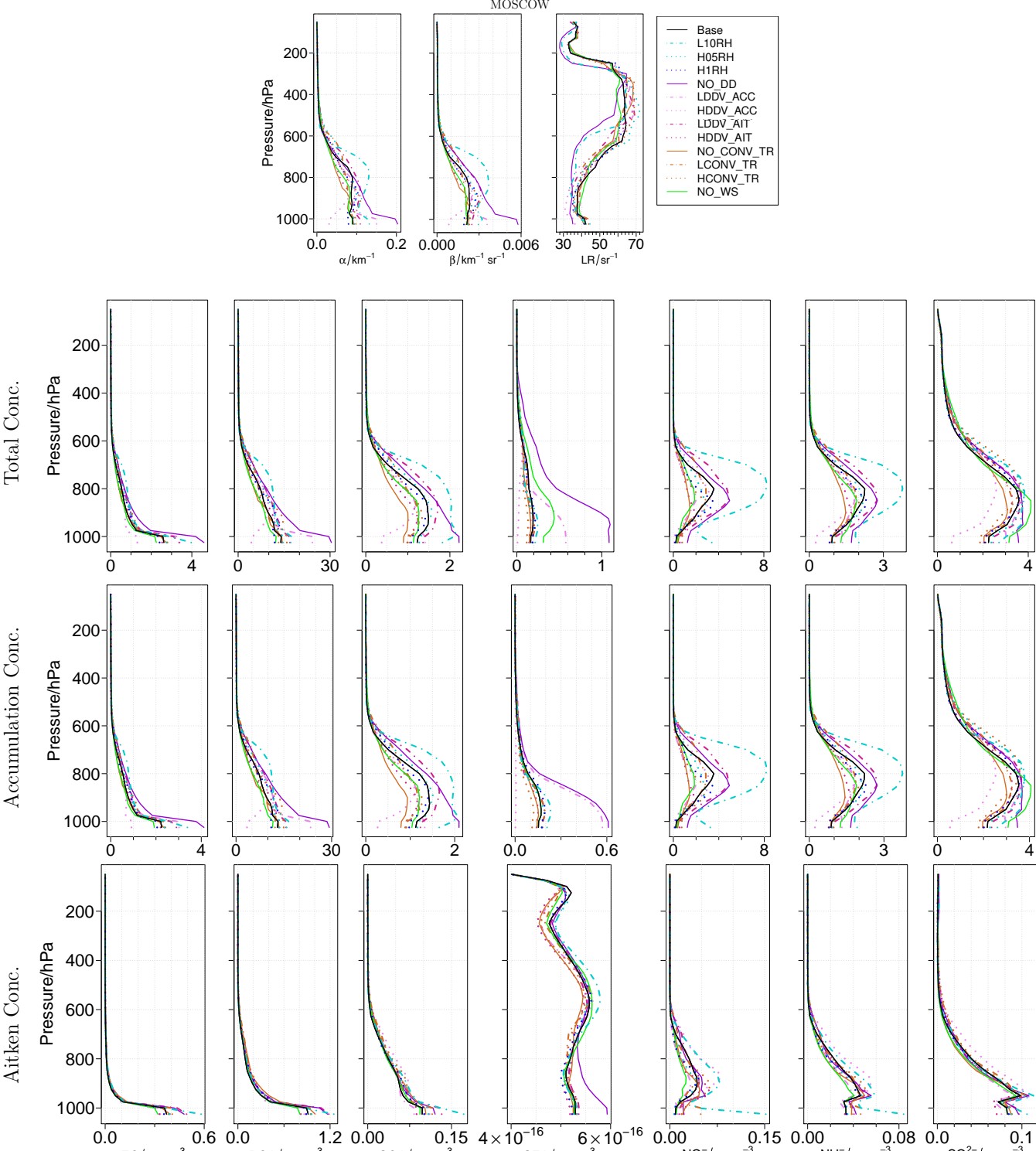

**Figure 7.** As Figure 3 but over the Moscow location.

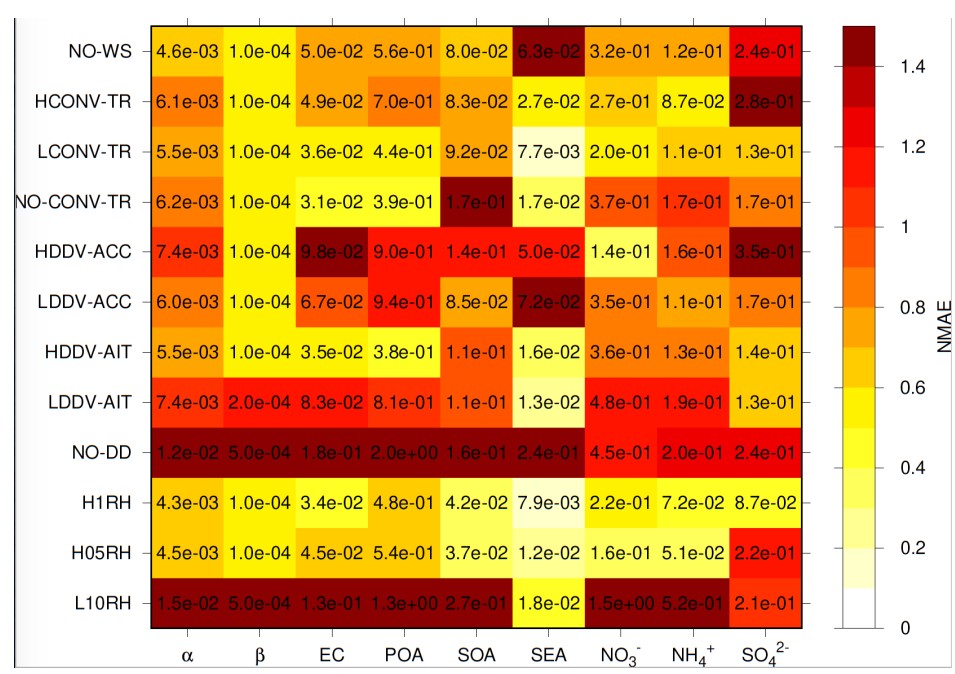

**Figure 8.** As Figure 4 but over Moscow location.