# Peer review of "Quantifying the sensitivity of aerosol optical properties to the parameterizations of physico-chemical processes during the 2010 Russian wildfires and heatwave"

_Atmospheric Chemistry and Physics, 2020_

## Referee Comment (RC1) · Anonymous Referee #1 · 30 Mar 2020

Palacios-Peña and co-authors have analyzed the sensitivity of simulated aerosol optical properties (aerosol optical depth, extinction and backscatter) to parameterizations of several physico-chemical processes. They did the analysis using the fully-coupled on-line WRF-Chem model and concentrating on the well documented Russian wildfire event in 2010. Their sensitivity studies showed some non-linear responses in optical properties which is a surprising result. Therefore, the manuscript is relevant to the readers of the ACP and it will likely be interesting to a wide audience. The methods are sound and described in enough detail, and the assumptions appear to be reasonable.

[Figure]

My detailed comments are given below.

My main concern with this study is related to the comparability of the different sensitivity tests. For example, RH is changed by -10 %, +0.5 %, and +1 %, whereas dry deposition velocity is scaled by 0.5 and 2 for the Aitken mode and 0.1 and 10 for the accumulation mode. As these changes vary from 0.5 % to 1000 % it is quite hard to understand how these sensitivity tests compare with each other. And why didn't you simulate RH reduction of 1 % and 0.5 %? Then they could have been compared directly with the enhancements. I think, it would be good to explain in the text why these changes are thought to be representative, meaningful and comparable for the parameters. For example, do they represent similar portions of the total ranges of the parameters? Or do they map the uncertainty ranges of these parameters? In order to say that sensitivity of the optical properties to RH is more important than to dry deposition, the changes in the parameters should be somehow comparable. This could be the true for current the analysis but it is not clear to the reader.

As the sensitivity test were made for an exceptional event, it is not clear how easily they can be generalized. For example, the low amount of clouds around the fire regions makes it seem that wet scavenging is not that important. However, based on Figure 2, wet scavenging has the strongest impact in the low-AOD region as there are more clouds present. Could you please clarify in the text if the conclusions are limited to specific conditions or if there are other processes which might have a stronger effect in some conditions.

I would have also liked to see a bit more detailed discussion on the significance of these results. Lately, there have been some studies where identical anthropogenic aerosol fields have been used in different models. For example, Nordling et al. (2019) found significant differences in the aerosol forcing between the models and they concluded that differences in model circulation responses appear to dominate the differences in regional climate responses. So, I feel that it would be an interesting addition to discuss (and compare at some level) the significance of the processes analyzed in this

manuscript and uncertainties in simulated circulation.

Page 1, lines 10-11: Are these absolute or relative differences. Would be good to clarify.

Page 2, line 23: "larger uncertainty", larger than what? Please, clarify.

Page 2, line 33: "as aerosol optical properties" → such as aerosol optical properties. Can optical properties be considered as a process?

Page 2, line 47: "high" → highly

Page 3, line 80: "To achieve this objective", it is not entirely clear what is objective you are referring to.

Page 3, line 83: What are the wavelengths of the AOD, extinction and backscatter coefficients? Please, mention them in the text.

Page 4, line 89: This sentence is a bit confusing: " with monthly mean temperatures in the summer months 5–9âŮęC higher than those for 2002–2009 due to a prolonged blocking anticyclone situation which triggered large wildfires". First of all, I'm not sure what you mean with the temperature comparison. Were the monthly mean temperatures 5-9 degrees warmer than the monthly averages for 2002-2009 or was the comparison done for individual months and the temperature range covers all these months? Secondly, I don't think it was the anticyclone situation which triggered the fires. I believe it was the people and the meteorological situation just made the thing worse.

Page 4, line 107: I wouldn't call 0.95 a very high single-scattering albedo as sulphate aerosols have SSA close to unity. Or do you mean that the SSA was high for smoke aerosols?

Page 4, line 116: There seems to be something missing from the end of the sentence ("in the "). Also the unit should be Wm-2.

Page 5, line 142: Temporal profile of what? It seems that this sentence is missing

some words.

Page 6, line 156: "as nucleation, chemistry or uptake of water" → such as nucleation, chemistry and uptake of water

Page 7, line 193: Is the modelled AOD in Figure 2 an average over the studied period? I'm just wondering if an average is the best way to present the data as there was a lot of variability in AOD during the episode and single outliers can have a big impact on averages. Did you check how the results would look if you would use medians instead of averages? It would also be interesting to see the variability of AOD during the period. It is likely higher in the MAX-AOD and Moscow points than in the MIN-AOD point, which might have an effect on the differences between the studied points.

Page 7, line 194: "The top-right figure shows the mean bias", do you mean the text in the top-left corner of the plots?

Page 7, line 208; "but less significant", compared to what? I'm surprised that there isn't more discussion on the HDDV_ACC simulation as it produces the largest mean change in AOD (-0.06).

Page 7, line 213 and 216: The indices of the subplots seem to have been mixed: Figure 2,j → Figure 2,k, Figure 2,k → Figure 2,j

Page 8, line 217: What do you mean with smooth differences? Please clarify.

Page 8, line 220: Here you could also mention that the mean AOD difference (0.04) for this simulation is the second largest even though there aren't many clouds in the studied domain. It implies that the wet scavenging is really important when there are clouds present.

Page 8, line 224: "where the spot where" → the spots where. "claims to bring" → aims to bring

Page 8, line 225: "time mean" → temporal mean

Page 8, line 228: Please clarify in the text that these are profiles are temporal averages. Would the results look the same if medians were used instead of averages?

Page 8, line 230: "$\alpha$ and $\beta$ present similar profiles" sounds a bit strange to me. Do you mean that the profiles of $\alpha$ and $\beta$ have similar shapes?

Page 8, line 232: This and the following sentence are missing "for $\beta$" after the backscatter values.

Page 8, line 245: These low LR values are a surprising and interesting result. Especially, as the LR values over the MIN-AOD location are in the same range as reported by Mielonen et al. (2013). What could explain the large difference in the source and reasonable results farther away? I believe this would be an interesting point to discuss in the manuscript.

Page 9, line 251: Please, clarify in the text how you calculated the mean absolute error for the profiles in practice. Did you first calculate the errors for each model level and then average them for the whole profile?

Page 9, line 252: I didn't quite catch how you calculated the normalized error. Could you please clarify? Which values were used in the normalization and how was it done in practice? Did you use the pixel-wise mean values from the base case at each model level or averaged over the whole column?

Page 9, line 262: "optical properties profiles" → profiles of extinction and backscatter coefficients

Page 9, line 277: "However, in the case with a reduction of the RH in a 10 % (L10RH), $NO-3$ displays a similar concentration as the base caseat surface levels and higher at levels above 800 hPa.", to me it seemed that the concentrations were similar only at surface levels and around 800 hPa. Did I read the figure wrong?

Page 10, line 288: "hidroxy" → hydroxy

[Figure]

Page 10, line 294: "the shape of the NOx and SOA profiles are similar, and thus, at these vertical levels, variations in SOA concentrations may be due to the effect described by Sarrafzadeh et al. (2016): an increase in NOx concentrations at low-NOx conditions (less than 30 ppb or around 55 $\mu$g m$-3$)", this is a bit hard to follow. Would something like this work better: the shape of the NOx and SOA profiles are similar, and thus, at these vertical levels, variations in SOA concentrations may be due to an increase in NOx concentrations at low-NOx conditions (less than 30 ppb or around 55$\mu$g m$-3$; Sarrafzadeh et al. (2016))

Page 10, line 297: "meanwhile in the L10RH case the positive variation of the concentration of SOA caused by the RH is limited.", I'm not sure what you mean with this. Could you please clarify?

Page 10, line 300: "provokes" → provoke

Page 10, line 309: With "target area" you mean the MAX-AOD point? I find it interesting that in this NO_DD simulation the positive AOD change forms a similar arch as in the H05RH simulation whereas other simulation exhibit a more uniform blob around the MAX-AOD point (see Figure 2). Furthermore, the lowest values in the NO_WS simulation match approximately the "gap" in the blobs of the NO_DD and H05RH simulations. Could some specific process explain these common features in these simulations? Also, in this NO_DD simulation the AOD increased a lot more around the MAX-AOD point so would the conclusions have changed if the selected point would have been a bit more eastward (see Figure 2)? Based on the AOD changes shown in Figure 2, moving the point slightly eastwards would not affect the magnitude of the change much in most simulations. Maybe doing the profile analysis with averages calculated over a number of pixels would give more robust results as the AOD changes in all the simulation are not smooth around the MAX-AOD and Moscow points?

Page 10, line 315: Please note that Supplementary Figures 2 and 3 are not mentioned in the text at all.

[Figure]

Page 10, line 316: "modifying the accumulation mode" → modifying the deposition of the accumulation mode. And the same change for the Aitken mode on the next line.

Page 11, line 327: "However, those species which are not directly emitted but are products of atmospheric chemistry (secondary aerosols), as SOA (NMAE>0.8 and MAE 0.2283) and most of the secondary inorganic species have their concentrations peak higher than thosein the base case between 900 and 600 hPa", this sentence is hard to follow. Please, revise.

Page 11, line 329: "fires area" → fire area, "optical properties profiles" → profiles of optical properties

Page 11, line 333: The Greenfield gap may not be familiar to all readers so, please, provide a size range and a reference for it.

Page 11, line 340: "When the profiles are analyzed, the response differs between species. EC, POA and $NO_3^-$ shows a slight reduction in their concentration, and SOA exhibits a large reduction.", is this correct? Based on Figure 3, it seemed that the concentrations of $NO_3^-$ and SOA increased. Furhermore, the SEA concentration appeared to decrease and not increase as mentioned in the text and the the highest SO2-4 concentrations appear to be around 800 hPa, not near the surface. Did I read the figure correctly?

Page 11, line 348: "$\beta$ profile is similar to the profiles of organic species (EC, POA and SOA)", to me it seems that the $\beta$ profile is also similar to NH-4 and SO2-4 profiles.

Page 12, line 358: "scaled to 1.5" → scaled by 1.5

Page 12, line 359: "For these species, the NO_CONV_TR experiment exhibits a concentration profile similar to the base case with slightly higher concentrations at surface levels and lower at higher levels", isn't it the opposite for the SEA concentrations? And the SOA, NO-3, and NH-4 concentrations appear to be constantly smaller than the base case? It would also be good to mention at the beginning of each section that

which point is analyzed. I'm guessing this analysis is related to the MAX-AOD point.

Page 12, line 369: "'differantly as" → differently than

Page 12, line 376: "show a peak in their profiles around the PBL" → show a peak around the PBL

Page 12, line 382: "at surface levels" → at the lowest levels

Page 12, line 383: "below the PBL" → below 800 hPa

Page 12, line 384: "highlights the high impact of organic species", please, clarify this statement. The concentrations of inorganics over PBL are also decreased so why are organics are more important? Is it related to the higher mass of POA (max 150 $\mu$g m-3 vs. ~4 $\mu$g m-3)

Page 13, line 387: "experiment is that with the strongest" → experiment has the strongest

Page 13, line 391: "It should also be highlighted that over the MIN-AOD and Moscow spots, EC and POA profiles of the assessed experiments show larger differences between them than over the MAX-AOD. This fact could be explained because over these locations these species are not being directly emitted. Moreover, the farther the location is, the larger the differences are." This statement could be true in relative sense but what about in absolute values? The concentration scales in the figures for the different locations are quite different. For example, the POA concentration scale is up to 150 $\mu$m m-3 at MAX-AOD, 30 $\mu$m m-3 at Moscow and only 2 $\mu$m m-3 at MIN-AOD. Therefore, based on figures 3, 5, 7 it is quite impossible to say which location has the largest changes in absolute sense. Could you please discuss this in more detail in the text?

Page 13, line 397: "In order to reduce", please explain how the uncertainties can be reduced based on the results presented in this study.

[Figure]

Page 13, line 400: "carried out during" → carried out for

Page 13, line 412: What do you mean with "important supersaturation"?

Page 14, line 442: It would be good to mention in this paragraph that the simulated LR values were different only/mainly at the MAX-AOD location, not everywhere.

Figure 2: Please, consider using binned color scale for the base AOD plot as well. Currently, the different hues are quite hard to differentiate. A binned color scale would make it easier to see the differences between the regions.

Figures 3, 5, and 7: Currently the lines are quite hard separate from each other. Maybe thicker lines would make it easier to see the colors?

Figures 4, 6, 8: Please, consider using binned color scale for the NMAE as it could make it easier to compare the different cases.

References

Nordling, K., Korhonen, H., Räisänen, P., Alper, M. E., Uotila, P., O'Donnell, D., and Merikanto, J.: Role of climate model dynamics in estimated climate responses to anthropogenic aerosols, Atmos. Chem. Phys., 19, 9969–9987, https://doi.org/10.5194/acp-19-9969-2019, 2019.

---

## Referee Comment (RC2) · Anonymous Referee #2 · 3 Apr 2020

**Comment on "Quantifying the sensitivity of aerosol optical properties to the parameterizations of physico-chemical processes during the 2010 Russian wildfires and heatwave " by Laura Palacios-Peña et al. (2020)**

**Summary/recommendation:**

This paper focuses on a studying the impact of key processes on aerosol optical properties representation. To that end, the authors conducted different sensitivity tests, modifying dry deposition, convective transport, relative humidity and wet scavenging during the 2010 Russian heatwave/wildfires episode using WRF-CHEM. They showed that changes in these key processes (mainly dry deposition, convective transport and relative humidity ) lead to changes in the sulphate-nitrate-SOA formation and thus to larger impact on AOD representation.

The scientific approach is sound and the work presented is substantial. However, the conclusions and discussions deserve more work before publication. The risk is that the impact of the publication to a broader scientific community remains limited unless the authors put the conclusions into a wider perspective.

I do recommend the publication of the current manuscript in ACP journal if the authors consider these minor revisions in order to make a nice addition to the literature. I request that the authors consider the following points as they revise this manuscript:

**General comments:**

1/ In this paper, the authors chose to focus on the 2010 Russian heatwave/wildfires episode. However, I would have also liked to see a section dedicated to a scientific discussion including more references to previous works on the subject aiming other simulation periods, other regions affected by wildfires ... in order to have wider conclusions and to highlight the significance of these results.

2/ The authors mentioned (only in the conclusion) that other processes (not discussed in their work) may also impact the aerosol optical properties representation. I believe that the paper could be further strengthened by adding a section in which the authors can compare their findings to more references that also discussed and analyzed the sensitivity of aerosol properties to other crucial parameters (such as, aerosol mixing state).

**Specific comments:**

1/ Page 1, lines 10 -11: Please clarify if these differences are absolute or relative.

2/ Page 3, lines 74-76: "The sensitivity tests were carried out using the WRF-Chem regional fully-coupled model by modifying dry deposition, sub-grid convective transport, relative humidity and wet scavenging." This sentence is repeated twice in the paper (here and in the abstract). Please formulate in a different way.

3/Page 3, line 83: Please add the wavelengths at which the aerosol optical properties (AOD, extinction and backscatter coefficients) are calculated.

4/ Page 5, lines 142-144: How these fire emissions are taken into account in the model? Can the authors give a brief description of the inventory and the uncertainties of the fire emissions used in this work.

5/ Page 6, section 2.3 : I think that the authors should better have two different sections: a section where they explain why they chose these "key sources" and another section where they describe the different sensitivity tests considered in their study.

6/ Page 7, line 208; "... showed the strongest response located over the wildfires area, but less significant.", what do you mean here by "less significant"?

7/ Page 7, line 194: "The top-right figure shows the mean bias". Does the top-right figure in figure 2 shows the mean relative differences or the mean bias between experiments and the base case? Please clarify.

8/ Page 8, line 229: What are these "SOA"?

9/ Page8, lines 228-229: Why did the authors evaluate only these species concentrations?

10/ Page 9, lines 251-252: How did the authors calculated the mean absolute error for the profiles ? Can the authors add a definition of the statistical indicators used in their sensitivity study?

11/ Page 10, line 315: Figures 2 and 3 in the supplementary material are not described or used in the text at all. Please add them.

12/ Page 11, line 333: What is the "Greenfield gap"? Please explain and add a reference.

13/ Page 13, line 397: "In order to reduce (or, at least, quantify) this uncertainty ..." How can we use the findings of this paper to reduce uncertainties? Please explain.

14/ Page 13, line 409-410: Can the authors give more details about these papers' findings in order to highlight these similarities ?

15/ Page 14, line 422: what are these VOC ?

**Technical comments:**

1/ Page 1, line 6: Please add a comma, after " In order to achieve this objective sensitivity".

2/ Page 1, line 7 and Page 5, line 122: Please replace "fully coupled" by "fully-coupled".

3/Page 2, line 23: Please correct "larger uncertainty" by "large uncertainty".

4/ Page 2, line 47: "Please correct "high influenced" by "highly influenced".

5/ Page 5, line 141: Please correct ($PM_{10}$ ..).

6/ Page 7, line 211: " provoke" please correct.

7/Page 8, line 225: Please replace "time mean" by "temporal mean".

8/ Page 10, line 288: "hydroxyl radical" please correct.

9/ Page 10, line 300: "does not provoke" please correct.

10/ Page 12, line 377: Please add a comma after " For the species, ... "

11/ Page 16, References section: in the ACP journal, the name of the journals should be cited in abbreviations. Please correct.

12/ Page 24, Figure 1: for the clearness of the figure, please fill the box (for the fire-affected target area ) with a more transparent color.

---

## Author Response (AR1)

*Dear Editor, Atmospheric, Chemistry and Physics Discussion:*

*Please find below our item-by-item response to the Reviewer's comments regarding manuscript **"Quantifying the sensitivity of aerosol optical properties to the parameterizations of physico-chemical processes during the 2010 Russian wildfires and heatwave"** by L. Palacios-Peña et al.*

*Do not hesitate to contact us with further questions.*

*With kind regards,*

*Laura Palacios Peña*

*First of all, we would gratefully thank all the Editor and Reviewers for their valuable comments on the manuscript.*

Anonymous Referee #1:

*Q: My main concern with this study is related to the comparability of the different sensitivity tests. For example, RH is changed by -10 %, +0.5 %, and +1 %, whereas dry deposition velocity is scaled by 0.5 and 2 for the Aitken mode and 0.1 and 10 for the accumulation mode. As these changes vary from 0.5 % to 1000 % it is quite hard to understand how these sensitivity tests compare with each other. And why didn't you simulate RH reduction of 1 % and 0.5 %? Then they could have been compared directly with the enhancements. I think, it would be good to explain in the text why these changes are thought to be representative, meaningful and comparable for the parameters. For example, do they represent similar portions of the total ranges of the parameters? Or do they map the uncertainty ranges of these parameters? In order to say that sensitivity of the optical properties to RH is more important than to dry deposition, the changes in the parameters should be somehow comparable. This could be the true for current the analysis but it is not clear to the reader.*

*A: Following both reviewer suggestions, the section 2.3 "Sensitivity test" has revised and the explanation of the reasons for the selection of the ranges of the parameters has been expanded.*

*"RH […] In order to avoid unlikely supersaturation values (higher than 1%) the chosen upper values were 1.005 and 1.01; that is, 0.5% and 1% supersaturation respectively. However, these variations would be irrelevant in the opposite direction (-0.5 and -1%). Because of that and following the evaluation of this meteorological variable conducted by Tucella et al. (2012) and Zabkar et al. (2015), this variable was scaled to 0.9 (a reduction of 10%).*

*[…] The modification for our sensitivity test regarding dry deposition consists on scaled DDV by the values indicated in Table~1. Following Lee et al. (2013), DDV has been scaled to 0.5 and 2 for the Aitken mode and 0.1 and 10 for the Accumulation mode, which are the both ends of the uncertainty range of these parameters. […].*

*[…] sub-grid convective transport […] Following the evaluations carried out by Doherty et al. (2005) and Quan et al. (2016), the output temporal tendency has been scaled to ±50%. […]"*

*The following Table indicates the responses of the different simulations to the variations specified. As seen on the Table, the sensitivity experiments have been selected so that they lead to analogous maximum modifications of AOD.*

*Table 1. Key processes of the sensitivity tests and their variation. The maximum AOD response to these variations is indicated in the last column.*

| Process | Sensitivity variation | Relative sensitivity variation | MAX AOD variation |
|---|---|---|---|
| L10RH | *0,9 | 10% | 0,6 |
| H05RH | *1,05 | 0.5% | < 0,15 |
| H1RH | *1,1 | 1% | 0,2 |
| NO_DD | OFF | 100% | 0,35 |
| LDDV_AIT | *0,5 | 50% | 0,3 |
| HDDV_AIT | *2 | 100% | 0,3 |
| LDDV_ACC | *0,1 | 90% | 0,3 |
| HDDV_ACC | *10 | 1000% | 0,3 |
| NO_CONV_TR | OFF | 100% | 0,4 |
| HCONV_TR | *0,5 | 50% | 0,4 |
| LCONV_TR | *1,5 | 50% | 0,4 |
| NO_WS | OFF | 100% | 0,2 |

*Q: […] Could you please clarify in the text if the conclusions are limited to specific conditions or if there are other processes which might have a stronger effect in some conditions.*

*A: A new section has been included where the limitations of this study are discussed highlighting other sources of errors and the extent of the conclusions due to focus only one episode. More details are explained in the next reply.*

*Q: I would have also liked to see a bit more detailed discussion on the significance of these results. Lately, there have been some studies where identical anthropogenic aerosol fields have been used in different models. For example, Nordling et al. (2019) found significant differences in the aerosol*

*forcing between the models and they concluded that differences in model circulation responses appear to dominate the differences in regional climate responses. So, I feel that it would be an interesting addition to discuss (and compare at some level) the significance of the processes analyzed in this manuscript and uncertainties in simulated circulation.*

*A: As both reviewers suggested a new section with an extensive discussion of the results regarding other processes, regions, periods and/ or conditions has been included in the manuscript.*

[revised manuscript text omitted]

*Q: Page 1, lines 10-11: Are these absolute or relative differences. Would be good to clarify.*

*A: They are absolute differences. This has been clarified in the text.*

*Q: Page 2, line 23: "larger uncertainty", larger than what? Please, clarify.*

*A: This uncertainty is larger than for any other climate forcing agents. This has been clarified in the text. […] "one of the forcing agents with the largest uncertainty in the climate system"*

*Q: Page 2, line 33: "as aerosol optical properties" → such as aerosol optical properties. Can optical properties be considered as a process?*

*A: The reviewer is right; this sentence could lead to a misunderstanding. Because of that, the sentence has been rewritten as follow: "Numerical models are useful tools for understanding the different parameters influencing the atmospheric system, such as aerosol optical properties."*

*Q: Page 2, line 47: "high" → highly. A: Corrected*

*Q: Page 3, line 80: "To achieve this objective", it is not entirely clear what is objective you are referring to. A: This sentence has been rewritten in order to clarify the text.*

*Q: Page 3, line 83: What are the wavelengths of the AOD, extinction and backscatter coefficients? Please, mention them in the text.*

*A: Added*

*Q: Page 4, line 89: This sentence is a bit confusing: "with monthly mean temperatures in the summer months 5–9°C higher than those for 2002–2009 due to a prolonged blocking anticyclone situation which triggered large wildfires". First of all, I'm not sure what you mean with the temperature comparison. Were the monthly mean temperatures 5-9 degrees warmer than the monthly averages for 2002-2009 or was the comparison done for individual months and the temperature range covers all these months? Secondly, I don't think it was the anticyclone situation which triggered the fires. I believe it was the people and the meteorological situation just made the thing worse.*

*A: We mean that the monthly mean temperatures 5-9 degrees warmer than the monthly averages for 2002-2009. In order to clarify the meaning, this sentence has been rewritten as follow: "with a prolonged blocking anticyclone situation which favored an increase of the summer temperature (close to 9 degrees larger than 2002-2009 summers) promoting to larger wildfires"*

*Q: Page 4, line 107: I wouldn't call 0.95 a very high single-scattering albedo as sulphate aerosols have SSA close to unity. Or do you mean that the SSA was high for smoke aerosols?*

*A: The adjective very high has been removal in order to avoid misunderstandings.*

*Q: Page 4, line 116: There seems to be something missing from the end of the sentence ("in the"). Also, the unit should be Wm-2.*

*A: The reviewer was right. Both typos have been corrected.*

*Q: Page 5, line 142: Temporal profile of what? It seems that this sentence is missing some words.*

*A: Temporal profile of emission. The sentence has been rewritten in order to clarify this point.*

*Q: Page 6, line 156: "as nucleation, chemistry or uptake of water" → such as nucleation, chemistry and uptake of water*

*A: Corrected*

*Q: Page 7, line 193: Is the modelled AOD in Figure 2 an average over the studied period? I'm just wondering if an average is the best way to present the data as there was a lot of variability in AOD during the episode and single outliers can have a big impact on averages. Did you check how the results would look if you would use medians instead of averages? It would also be interesting to see the variability of AOD during the period. It is likely higher in the MAX-AOD and Moscow points than in the MIN-AOD point, which might have an effect on the differences between the studied points.*

*A:*

[Figure]

*Figure 1. Temporal mean of modelled AOD at 550 nm for the base case (top-left) and mean bias differences between experiments and the base case.*

[Figure]

Figure 2. *Temporal median of modelled AOD at 550 nm for the base case (top-left) and median bias differences between experiments and the base case*

These figures show the temporal mean and the differences (top) and the temporal median and the temporal median of the differences (bottom). Figures reveal that for both, AOD and differences, the use of the temporal median displays similar results but less intense.

[Figure]

Figure 3. Temporal variability of modelled AOD at 550 nm for the base case (top-left) and temporal variability of the differences between experiments and the base case

*Temporal variability is shown in the above figure. The reviewer suggested that the largest differences over the MAX-AOD locations are due to its high temporal variability meanwhile the small differences over the MIN-AOD location are due to its small temporal variability. However, according to the figure, temporal variability over the MOSCOW and MIN-AOD locations is similar meanwhile our results show higher impacts of the sensitivity test over the MOSCOW location.*

Q: Page 7, line 194: "The top-right figure shows the mean bias ", do you mean the text in the top-left corner of the plots?

*A: This sentence has been removed. This was a wrong sentence from an older version of the manuscript.*

Q: Page 7, line 208; "but less significant", compared to what? I'm surprised that there isn't more discussion on the HDDV_ACC simulation as it produces the largest mean change in AOD (-0.06).

*A: This sentence has been rewritten for the sake of clarification: "All the experiments related to changes in dry deposition (Figure 2,d-h) showed its strongest response located over the wildfires area, but this response is less relevant than for other cases." Moreover, a brief discussion of the HDDV_ACC test has been included: "HDDV_ACC is the only test which produces a general reduction in AOD over most of the study area (temporal and spatial mean change in AOD of -0.06) but this differences are stronger over fire affected areas and downwind."*

Q: Page 7, line 213 and 216: The indices of the subplots seem to have been mixed: Figure 2, j → Figure 2, k, Figure 2, k → Figure 2, j

*A: Corrected*

Q: Page 8, line 217: What do you mean with smooth differences? Please clarify.

*A: What smooth differences means is that differences in the LCONV_TRANS are lower in absolute terms. We try to clarify this point with this new sentence: "Figure 2,j indicates that the Low sub-grid Convective Transport case (LCONV_TR) has lower absolute differences are lower."*

Q: Page 8, line 220: Here you could also mention that the mean AOD difference (0.04) for this simulation is the second largest even though there aren't many clouds in the studied domain. It implies that the wet scavenging is really important when there are clouds present.

*A: We thanks the reviewer for its valuable comment. This has been included in the section 3.1.*

Q: Page 8, line 224: "where the spot where" → the spots where. "claims to bring" → aims to bring

*A: Corrected*

Q: Page 8, line 225: "time mean" → temporal mean

*A: Corrected*

Q: Page 8, line 228: Please clarify in the text that these are profiles are temporal averages. Would the results look the same if medians were used instead of averages?

*A: This has been clarified in the text. Regarding the use of the medians, the median of modelled AOD for the base case is shown above. This figure indicates that median AOD values are a bit lower that mean AOD values. However, the spatial pattern is similar for both, median and mean. Consequently, using the median instead of the mean does not change the conclusion obtained for the vertical profiles.*

Q: Page 8, line 230: "$\alpha$ and $\beta$ present similar profiles" sounds a bit strange to me. Do you mean that the profiles of "$\alpha$ and $\beta$ have similar shapes?

*A: The reviewer is right. This has been corrected*

Q: Page 8, line 232: This and the following sentence are missing "for $\beta$" after the backscatter values.

*A: Added*

Q: Page 8, line 245: These low LR values are a surprising and interesting result. Especially, as the LR values over the MIN-AOD location are in the same range as reported by Mielonen et al. (2013). What could explain the large difference in the source and reasonable results farther away? I believe this would be an interesting point to discuss in the manuscript.

*A: LR values over the MIN-AOD location are not comparable to those values reported by Mielonen et al. (2013) because these latter were reported over biomass burning affected areas. Over the MIN-AOD location, sea salt is predominant. Over this location LR values expected should be close to 30 sr$^{-1}$. Moreover, extinction and backscatter modelling profiles shape are similar, which*

*is not similar to most of the observed profiles, resulting in a misrepresentation of the LR. This could be ascribed to the model estimation of these aerosol optical properties profiles.*

*This clarification has been included in the text. Page 9, line 256: "[…] It is noticeable that LR values over the MIN-AOD location (close to 30 sr$^{-1}$) are not comparable to those values expected by the scientific literature (e.g. Mielonen et al., 2013). However, it should be born in mind that MIN-AOD location is affected principally by sea salt, while the aforementioned reference covers a biomass-burning affected area. Moreover, extinction and backscatter modelling profiles shape are similar (rather constant at levels close to the surface), which is not found in most of the observed LR profiles. This could be ascribed to a model misrepresentation of extinction and backscatter modelling profiles. […]"*

*Q: Page 9, line 251: Please, clarify in the text how you calculated the mean absolute error for the profiles in practice. Did you first calculate the errors for each model level and then average them for the whole profile?*

*Page 9, line 252: I didn't quite catch how you calculated the normalized error. Could you please clarify? Which values were used in the normalization and how was it done in practice? Did you use the pixel-wise mean values from the base case at each model level or averaged over the whole column?*

*A: Both, the mean absolute error (MAE) and the normalize MAE (NMAE) were estimated by computing the error at each model level and then averaging along the whole profile. Regarding NMAE, as indicated in the manuscript, it was normalized dividing by the base case at each level. The objective was to show the magnitude of the relative changes in each sensitivity test for each evaluated variable. The two statistics were computed as follows:*

$$MAE = \frac{\sum_1^n |x_{test} - x_{base}|}{n}$$

$$NMAE = \frac{\sum_1^n \frac{|x_{test} - x_{base}|}{x_{base}} x100}{n}$$

*where n is the number of levels and x is the evaluated magnitude for the sensitivity test ($x_{test}$) and the base experiment ($x_{base}$).*

*To clarify the estimation of these two statistics, the manuscript has been rewritten: "In addition, Figures 4, 6 and 8 quantify the mean absolute error (MAE) of each experiment with respect to the base case, and in colors, the normalized MAE (NMAE). MAE has been estimated by averaging the absolute error of each*

*experiment regarding the base case at each model level. NMAE is the absolute error divided by the base case at each level and then averaged along the column. The NMAE analysis illustrates the relative change of each magnitude and helps to the intercomparison between the sensitivity test."*

*Q: Page 9, line 262: "optical properties profiles" → profiles of extinction and backscatter coefficients*

*A: Corrected*

*Q: Page 9, line 277: "However, in the case with a reduction of the RH in a 10 % (L10RH), $NO^{-3}$ displays a similar concentration as the base case at surface levels and higher at levels above 800 hPa.", to me it seemed that the concentrations were similar only at surface levels and around 800 hPa. Did I read the figure wrong?*

*A: The reviewer is right, and this sentence has been rewritten for the sake of clarity. "However, in the case with a reduction of the RH in a 10% (L10RH), $NO^3$ displays a similar concentration as the base case at the surface level and around 800 hPa. Throughout the rest of the profile concentration is higher than in the base case but not as higher as in the H1RH case."*

*Q: Page 10, line 288: "hidroxy" → hydroxy*

*A: Corrected*

*Q: Page 10, line 294: "the shape of the NOx and SOA profiles are similar, and thus, at these vertical levels, variations in SOA concentrations may be due to the effect described by Sarrafzadeh et al. (2016): an increase in NOx concentrations at low-NOx conditions (less than 30 ppb or around 55 µg m−3)", this is a bit hard to follow. Would something like this work better: the shape of the NOx and SOA profiles are similar, and thus, at these vertical levels, variations in SOA concentrations may be due to an increase in NOx concentrations at low-NOx conditions (less than 30 ppb or around 55µg m−3; Sarrafzadeh et al. (2016))*

*A: The sentence has been rewritten as in the reviewer's suggestion.*

*Q: Page 10, line 297: "meanwhile in the L10RH case the positive variation of the concentration of SOA caused by the RH is limited.", I'm not sure what you mean with this. Could you please clarify?*

*A: As in the L10RH case there is not strong increase in $NO_x$ the variation in SOA concentration cannot be highly dependent of the oxidation process described below and dependency of RH modification is higher. This has been clarified in*

*the text as: "That means that this variation depends more of RH modifications than NO$_x$ oxidations".*

*Q: Page 10, line 300: "provokes" → provoke*

*A: Corrected*

*Q: Page 10, line 309: With "target area" you mean the MAX-AOD point? I find it interesting that in this NO_DD simulation the positive AOD change forms a similar arch as in the H05RH simulation whereas other simulations exhibit a more uniform blob around the MAX-AOD point (see Figure 2). Furthermore, the lowest values in the NO_WS simulation match approximately the "gap" in the blobs of the NO_DD and H05RH simulations. Could some specific process explain these common features in these simulations? Also, in this NO_DD simulation the AOD increased a lot more around the MAX-AOD point so would the conclusions have changed if the selected point would have been a bit more eastward (see Figure 2)? Based on the AOD changes shown in Figure 2, moving the point slightly eastwards would not affect the magnitude of the change much in most simulations. Maybe doing the profile analysis with averages calculated over a number of pixels would give more robust results as the AOD changes in all the simulation are not smooth around the MAX-AOD and Moscow points?*

*A: Yes, the target area means here the MAX-AOD location. This has been clarified in the text.*

*Regarding the blobs of the NO_DD and H05RH simulations, although the bias patterns of NO_DD and H05RH could share some similarities, it is difficult to attribute them to a specific process since the both present a low signal. The orography may play a relevant role. In some experiments, positive bias is concentrated on left of the Volga valley and negative bias is on the right. The arch seems to follow the river path shape, but it is difficult to extract a robust conclusion.*

[Figure]

Q: Page 10, line 315: Please note that Supplementary Figures 2 and 3 are not mentioned in the text at all.

A: Their references has been included in the text

Q: Page 10, line 316: "modifying the accumulation mode" → modifying the deposition of the accumulation mode. And the same change for the Aitken mode on the next line.

A: Added

Q: Page 11, line 327: "However, those species which are not directly emitted but are products of atmospheric chemistry (secondary aerosols), as SOA (NMAE>0.8 and MAE 0.2283) and most of the secondary inorganic species have their concentrations peak higher than those in the base case between 900 and 600 hPa", this sentence is hard to follow. Please, revise.

A: The sentence has been rewritten as follow: "However, secondary aerosol; which are not directly emitted and are products of atmospheric chemistry; such as SOA (NMAE>0.8 and MAE 0.2283) and most of the secondary inorganic species have their concentrations peak at a higher altitude than those in the base case between 900 and 600 hPa"

Q: Page 11, line 329: "fires area" → fire area, "optical properties profiles" → profiles of optical properties

A: Corrected

Q: Page 11, line 333: The Greenfield gap may not be familiar to all readers so, please, provide a size range and a reference for it.

A: As reviewer suggested this has been included in the text

Q: Page 11, line 340: "When the profiles are analyzed, the response differs between species. EC, POA and $NO^{-3}$ shows a slight reduction in their concentration, and SOA exhibits a large reduction.", is this correct? Based on Figure 3, it seemed that the concentrations of $NO^{-3}$ and SOA increased. Furthermore, the SEA concentration appeared to decrease and not increase as mentioned in the text and the highest $SO_2^{-4}$ concentrations appear to be around 800 hPa, not near the surface. Did I read the figure correctly?

A: There is an error in the legend of the figure and the colors are changed. This has been corrected in the new figures.

Q: Page 11, line 348: "$\beta$ profile is similar to the profiles of organic species (EC, POA and SOA)", to me it seems that the $\beta$ profile is also similar to $NH^{-4}$ and $SO2^{-4}$ profiles.

A: The reviewer is right. This sentence has been corrected.

Q: Page 12, line 358: "scaled to 1.5" → scaled by 1.5

A: Corrected

Q: Page 12, line 359: "For these species, the NO_CONV_TR experiment exhibits a concentration profile similar to the base case with slightly higher concentrations at surface levels and lower at higher levels", isn't it the opposite for the SEA concentrations? And the SOA, NO-3, and NH-4 concentrations appear to be constantly smaller than the base case? It would also be good to mention at the beginning of each section that which point is analyzed. I'm guessing this analysis is related to the MAX-AOD point.

A: The reviewer is right and this point has been clarified in the text as follows: "For POA, EC and SO4-2, the NO_CONV_TR experiment exhibits a concentration profile similar to the base case with slightly higher concentrations at surface levels and lower at higher levels. The opposite behavior is displayed

*by SEA concentrations. Moreover, SOA, NO-3 and NH-4 concentration are constantly smaller than the base case."*

*The reviewer is right again, and the analyzed point is the MAX-AOD point. The text has been reviewed to clarify this somewhere was necessary.*

Q: Page 12, line 369:" differantly as" → differently than

*A: Corrected*

Q: Page 12, line 376: "show a peak in their profiles around the PBL" → show a peak around the PBL

*A: Corrected*

Q: Page 12, line 382: "at surface levels" → at the lowest levels

*A: Corrected*

Q: Page 12, line 383: "below the PBL" → below 800 hPa

*A: Corrected*

Q: Page 12, line 384: "highlights the high impact of organic species", please, clarify this statement. The concentrations of inorganics over PBL are also decreased so why are organics are more important? Is it related to the higher mass of POA (max 150 µg m$^{-3}$ vs. ~4 µg m$^{-3}$)?

*A: This is because the shape of the vertical profile of aerosol optical properties is quite similar to the shape of EC and POA. However, as the reviewer stated there is a reduction in the concentration of inorganics species and SOA over this location which could also impacts optical properties.*

Q: Page 13, line 387: "experiment is that with the strongest" → experiment has the strongest

*A: Corrected*

Q: Page 13, line 391: "It should also be highlighted that over the MIN-AOD and Moscow spots, EC and POA profiles of the assessed experiments show larger differences between them than over the MAX-AOD. This fact could be explained because over these locations these species are not being directly emitted. Moreover, the farther the location is, the larger the differences are." This statement could be true in relative sense but what about in absolute values? The

concentration scales in the figures for the different locations are quite different. For example, the POA concentration scale is up to 150 µm m$^{-3}$ at MAX-AOD, 30 µm m$^{-3}$ at Moscow and only 2 µm m$^{-3}$ at MIN-AOD. Therefore, based on figures 3, 5, 7 it is quite impossible to say which location has the largest changes in absolute sense. Could you please discuss this in more detail in the text?

A: What we mean with these differences was differences between the profile shape. Thus, the farther the location is the more different is the shape of the vertical profiles and this could be highly influenced by the transport processes. In order to clarify this statement, the paragraph has been rewritten as follows: "It should also be highlighted, the profile shape of EC and POA over the MIN-AOD and Moscow show larger differences than over the MAX-AOD area for the different experiments. These differences in the shape profiles could be attributed to these species are not directly emitted over MIN-AOD and Moscow areas and transport processes could be influenced by vertical distribution. Moreover, the farther the location is, the more different the shape of the vertical profile is."

Q: Page 13, line 397: "In order to reduce", please explain how the uncertainties can be reduced based on the results presented in this study.

A: This sentence has been rewritten in order to a better explanation: "This work assesses the sensitivity of aerosol optical properties and the aerosol vertical distribution to key physical processes. To achieve this objective, sensitivity runs modifying RH, dry deposition, sub-grid convective transport and wet scavenging have been carried out for the 2010 Russian heatwave/wildfires episode with the WRF-Chem regional fully coupled model. Findings in this work would help to improve modelling aerosol representation giving some initial guidelines about what parameters could be misrepresented or are the most sensitive to the vertical mixing."

Q: Page 13, line 400: "carried out during" → carried out for

A: Corrected

Q: Page 13, line 412: What do you mean with "important supersaturation"?

A: Supersaturation above 1% because supersaturation rarely exceeds 1%–2% and this only can be in warm clouds (Devenish et al.2016) in particular vigorous convective clouds (Prabha et al. 2011). This has been clarified in the text.

Q: Page 14, line 442: It would be good to mention in this paragraph that the simulated LR values were different only/mainly at the MAX-AOD location, not everywhere.

A: This has been clarified in the text as follows: "Regarding the LR, simulated values of this variable are remarkably different from those observed in the scientific literature, mainly over fire affected areas."

Q: Figure 2: Please, consider using binned color scale for the base AOD plot as well. Currently, the different hues are quite hard to differentiate. A binned color scale would make it easier to see the differences between the regions.

A: The figure has been redone as reviewer's suggestions

Q: Figures 3, 5, and 7: Currently the lines are quite hard separate from each other. Maybe thicker lines would make it easier to see the colors?

A: The figures have been redrawn as reviewer's suggestions

Q: Figures 4, 6, 8: Please, consider using binned color scale for the NMAE as it could make it easier to compare the different cases.

A: The figures have been redrawn as reviewer's suggestions

Anonymous Referee #2:

General comments:

Q: 1/ In this paper, the authors chose to focus on the 2010 Russian heatwave/wildfires episode. However, I would have also liked to see a section dedicated to a scientific discussion including more references to previous works on the subject aiming other simulation periods, other regions affected by

*wildfires ... in order to have wider conclusions and to highlight the significance of these results.*

*2/ The authors mentioned (only in the conclusion) that other processes (not discussed in their work) may also impact the aerosol optical properties representation. I believe that the paper could be further strengthened by adding a section in which the authors can compare their findings to more references that also discussed and analyzed the sensitivity of aerosol properties to other crucial parameters (such as, aerosol mixing state).*

*A: As both reviewers suggested a new section with an extensive discussion of the results regarding other processes, regions, periods and/ or conditions has been included in the manuscript. This discussion is in the response to reviewer #1.*

**Specific comments:**

*Q: 1/ Page 1, lines 10 -11: Please clarify if these differences are absolute or relative.*

*A: Differences are absolute. It has been clarified in the revised version of the manuscript.*

*Q: 2/ Page 3, lines 74-76: "The sensitivity tests were carried out using the WRF-Chem regional fully-coupled model by modifying dry deposition, sub-grid convective transport, relative humidity and wet scavenging." This sentence is repeated twice in the paper (here and in the abstract). Please formulate in a different way.*

*A: The sentence has been rewritten as follow: "This quantification has been estimated by sensitivity tests carried out using the WRF-Chem regional fully-coupled model. Modified aerosol processes and parameters are dry deposition, sub-grid convective transport, relative humidity and wet scavenging."*

*Q: 3/ Page 3, line 83: Please add the wavelengths at which the aerosol optical properties (AOD, extinction and backscatter coefficients) are calculated.*

*A: Added*

*Q: 4/ Page 5, lines 142-144: How these fire emissions are taken into account in the model? Can the authors give a brief description of the inventory and the uncertainties of the fire emissions used in this work?*

*A: The description of the fire emissions has been expanded as reviewer suggested. Moreover, a reference of the evaluation of this fire emission inventory has been provided.*

*"Biomass burning emission data of the total PM emissions (daily data with a spatial resolution of 0.1°) were derived from the project IS4FIRES (Integrated monitoring and modelling system for wild-land fires; Sofiev et al., 2009). As described by Soares et al., 2015 emissions were calculated from a re-analysis of the fire radiative power from MODIS on-board of Aqua and Terra satellites; and calibration emission factors based on the comparison between observations and modelled data processed by the System for Integrated modeLing of Atmospheric coMposition (SILAM). Day and night vertical injection profiles were also provided. Finally, total PM emissions were speciated to WRF-Chem emission species following Andreae and Merlet (2001) and Wiedinmyer et al. (2011). No heat release due to the fires was considered. Uncertainties were estimated by Soares et al.,2015 with an overestimation in-average of 20–30% which could raise to about 50% in specific episodes. This impacts on total emissions likely come from under-stated injection height which can lead to overestimation of the near-surface concentration and reduction of elevated plumes; or a misinterpretation by MODIS of oil and gas flares and large industrial installation as fires. More details can be found in Soares et al.,2015."*

*Q: 5/ Page 6, section 2.3: I think that the authors should better have two different sections: a section where they explain why they chose these "key sources" and another section where they describe the different sensitivity tests considered in their study.*

*A: Section 2.3 covers the definition of the sensitivity tests conducted in this contribution, while the key sources of uncertainty are profusely detailed in the introduction.*

*Q: 6/ Page 7, line 208; "...showed the strongest response located over the wildfires area, but less significant.", what do you mean here by "less significant"?*

*A: This sentence has been rewritten for the sake of clarification: "All the experiments related to changes in dry deposition (Figure 2,d-h) showed its strongest response located over the wildfires area, but this response is less relevant than for other cases."*

*Q: 7/ Page 7, line 194: "The top-right figure shows the mean bias ". Does the top-right figure in figure 2 shows the mean relative differences or the mean bias between experiments and the base case? Please clarify.*

*A: Figure 2 displays the mean bias between experiments and the base case. There was an error in the caption of the figure which has been corrected. The text has been checked in order to avoid similar errors.*

Q: 8/ Page 8, line 229: What are these "SOA"?

*A: SOA in this chemical mechanism are composed by SOA Anthropogenic and Biogenic organic, both dry and in cloud.*

Q: 9/ Page 8, lines 228-229: Why did the authors evaluate only these species concentrations?

*A: EC, POA and SOA were selected due to their importance in a biomass burning episode. NO3-, NH4- and SO2- have been selected because they are the main inorganic species and those involved in the sulphate-ammonium-nitrate-water equilibrium simulated by the ISORROPIA mechanism. Finally, SEA has been selected as an example of natural aerosol with a small impact on this episode. Dust concentrations are negligible over the target domain.*

Q: 10/ Page 9, lines 251-252: How did the authors calculated the mean absolute error for the profiles? Can the authors add a definition of the statistical indicators used in their sensitivity study?

*A: As we indicated in the response to the reviewer #1, a clarification of the estimation of these statistics has been added in the text.*

Q: 11/ Page 10, line 315: Figures 2 and 3 in the supplementary material are not described or used in the text at all. Please add them.

*A: A reference for these figures has been included in the text*

Q: 12/ Page 11, line 333: What is the "Greenfield gap"? Please explain and add a reference.

*A: As reviewer suggested this has been included in the text.*

*"(particle radii of the range of 0.1–1 μm where Brownian motion is not large anymore and gravitational settling is not yet important; Greenfield 1957; Ladino, et al. 2011)"*

Q: 13/ Page 13, line 397: "In order to reduce (or, at least, quantify) this uncertainty ..." How can we use the findings of this paper to reduce uncertainties? Please explain.

*A: This sentence has been rewritten in order to a better explanation: "This work assesses the sensitivity of aerosol optical properties and the aerosol vertical distribution to key physical processes. To achieve this objective, sensitivity runs modifying RH, dry deposition, sub-grid convective transport and wet scavenging have been carried out for the 2010 Russian heatwave/wildfires episode with the WRF-Chem regional fully coupled model. Findings in this work would help to improve modelling aerosol representation giving some initial guidelines about what parameters could be misrepresented or are the most sensitive to the vertical mixing."*

*Q:* 14/ Page 13, line 409-410: Can the authors give more details about these papers' findings in order to highlight these similarities?

*A: Following both reviewers suggestion this part of the manuscript has been rewritten including a comparison of our results with other similar studies.*

*Q:* 15/ Page 14, line 422: what are these VOC?

*A: VOC refers to Volatile organic compounds as described in page 10, line:XXX*

Technical comments:

*Q:* 1/ Page 1, line 6: Please add a comma, after "In order to achieve this objective sensitivity".

*A: Corrected*

*Q:* 2/ Page 1, line 7 and Page 5, line 122: Please replace "fully coupled" by "fully-coupled".

*A: Replaced*

*Q:* 3/Page 2, line 23: Please correct "larger uncertainty" by "large uncertainty".

*A: Corrected*

*Q:* 4/ Page 2, line 47: "Please correct "high influenced" by "highly influenced".

*A: Corrected*

*Q:* 5/ Page 5, line 141: Please correct ($PM_{10}$ ...).

*A: Corrected*

*Q:* 6/ Page 7, line 211: "provoke" please correct.

*A: Corrected*

*Q:* 7/Page 8, line 225: Please replace "time mean" by "temporal mean".

*A: Corrected*

*Q:* 8/ Page 10, line 288: "hydroxyl radical" please correct.

*A: Corrected*

*Q:* 9/ Page 10, line 300: "does not provoke" please correct.

*A: Corrected*

*Q:* 10/ Page 12, line 377: Please add a comma after "For the species, ..."

*A: Corrected*

*Q:* 11/ Page 16, References section: in the ACP journal, the name of the journals should be cited in abbreviations. Please correct.

*A: The references section is automatically done by the Bibtex tool. I think this time of typos will be corrected during the edition process.*

*Q:* 12/ Page 24, Figure 1: for the clearness of the figure, please fill the box (for the fire-affected target area) with a more transparent color.

*A: Modified*

***References:***

[revised manuscript text omitted]

---

## Author Response (AR2)

*Dear Editor, Atmospheric, Chemistry and Physics Discussion:*

*Please find below our item-by-item response to the Reviewer's comments regarding manuscript* **"Quantifying the sensitivity of aerosol optical properties to the parameterizations of physico-chemical processes during the 2010 Russian wildfires and heatwave"** *by L. Palacios-Peña et al.*

*Do not hesitate to contact us with further questions.*

*With kind regards,*

*Laura Palacios Peña*

*First of all, we would gratefully thank the Editor and Reviewers for their valuable comments on the manuscript.*

Anonymous Referee #1:

*Q: The revised manuscript by Palacios-Peña et al. is significantly better than the first submission. The authors replied to my comments in a satisfactory manner and they revised the manuscript accordingly. […] Here are some examples of the places where the language needs the checked (line numbers from the manuscript with mark-up) and a couple of minor comments:*

*Line 224: "this" → the A: Corrected*

*Q: Line 230: "has lower absolute differences are lower" → has lower absolute differences A: Corrected*

*Q: Line 258: This discussion is still a bit confusing for me. The authors state that "It is noticeable that LR values over the MIN-AOD location (close to 30 sr −1 ) are not comparable to those values expected by the scientific literature are (e.g Mielonen et al., 2013)." However, in Figure 5., which presents the LR profile for the MIN-AOD location, most simulations show similar LRs (above 60 sr-1) as reported by Mielonen et al. (2013), only the NO_DD simulation shows low LR values (closer to 40 sr-1 than 30 sr-1) so to me it seems that the simulated LRs are too high for a region which is not heavily affected by the smoke. The authors should check this analysis and clarify the text accordingly.*

*A: The text has been rephrased for a better clarification of the comparison with Mielonen et al. (2013) reference.*

*Q: Line 277: "change of each magnitude", what do you mean with magnitude? Please, clarify. A: With each magnitude, we mean each studied variable. The sentence has been rewritten for clarification as "The NMAE analysis illustrates*

*the relative change of each specie and optical properties and helps to the intercomparison between the sensitivity test"*

*Q: Line 302: "not as higher" → not as high* A: Corrected

*Q: Line 406: "how a peak" → show a peak* A: Corrected

*Q: Line 425: "These differences in the shape profiles could be attributed to these species are not directly emitted over MIN-AOD and Moscow areas and transport processes could be influenced by vertical distribution." This is a very confusing sentence. Do you mean something like this: These differences in the shapes of the profiles could be attributed to species which are not directly emitted over the MIN-AOD and Moscow areas thus, the vertical distribution could be influenced by transport processes. A: The sentence has been corrected as reviewer suggestion.*

*Q: Line 428: This discussion section is a good addition, but the language needs to be checked thoroughly. Currently, the text is quite hard to follow. In addition, the authors should clarify what is the take-home-message of this discussion. Which processes cause the largest uncertainty in ARI and ACI estimates? There are a lot of processes with big uncertainties but which of them have the biggest impact on the evaluation of aerosol climate effects and/or which of them have the largest knowledge gaps? This is a huge topic to discuss and I don't mean that the authors should go through all the details and try to compare numbers which are not directly comparable but I would like to have a bit better context for these findings, especially on the magnitude of the impact these uncertainties cause to our estimates of AOD.*

*A: The language has been thoroughly checked in this section and rewritten as suggested by the reviewer. The language in the conclusion section has been also checked.*

*Q: Line 440: "influenced by the latter" or do you mean "influenced by the former"? A: The sentence has been corrected as reviewer suggestion.*

*Q: Line 450: What is this "CCN uncertainty"?*

*A: This was previously explained in the discussion. Line 441-443"*

[revised manuscript text omitted]

---

## Author Response (AR3)

*Dear Editor, Atmospheric, Chemistry and Physics Discussion:*

*Please find below our item-by-item response to the Reviewer's comments regarding manuscript "**Quantifying the sensitivity of aerosol optical properties to the parameterizations of physico-chemical processes during the 2010 Russian wildfires and heatwave**" by L. Palacios-Peña et al.*

*Do not hesitate to contact us with further questions.*

*With kind regards,*

*Laura Palacios Peña*

*First of all, we would gratefully thank the Editor for their valuable work.*

**Editor Decision: Publish subject to technical corrections (13 Jul 2020) by Jianzhong Ma**

*Comments to the Author:*

*Figure 2 needs to be revised, perhaps just turning it around 90 degree in the clockwise direction. It might not be good to look at a plot upside down.*

*A: Figures has been corrected following the reviewer's suggestion.*